

# Molecular characterization of gaseous and particulate oxygenated compounds at a remote site in Cape Corsica in the western Mediterranean basin.

Vincent Michoud[1,2], Elise Hallemans[1,2], Laura Chiappini[2,†], Eva Leoz-Garziandia[2], Aurélie Colomb[3], Sébastien Dusanter[4], Isabelle Fronval[4], François Gheusi[5], Jean-Luc Jaffrezo[6], Thierry Léonardis[4], Nadine Locoge[4], Nicolas Marchand[7], Stéphane Sauvage[4], Jean Sciare[8,9], Jean-François Doussin[1]

[1] LISA, UMR CNRS 7583, Université de Paris, Université Paris-Est-Créteil, Institut Pierre Simon Laplace (IPSL), Créteil, France

[2] Institut National de l'Environnement Industriel et des Risques, Verneuil-en-Halatte, France

[3] LaMP, CNRS UMR6016, Clermont Université, Université Blaise Pascal, Aubière, France

[4] IMT Lille Douai, Univ. Lille, SAGE - Département Sciences de l'Atmosphère et Génie de l'Environnement, 59000 Lille, France

[5] Laboratoire d'Aérologie, Université de Toulouse, CNRS, Toulouse, France

[6] Université Grenoble Alpes, CNRS, IRD, IGE, 38000 Grenoble, France

[7] Aix Marseille Univ, CNRS, LCE, Marseille, 13003, France

[8] LSCE, CNRS-CEA-UVSQ, IPSL, Université Paris-Saclay, Gif-sur-Yvette, France

[9] EEWRC, The Cyprus Institute, Nicosia, Cyprus

† deceased

## Abstract

The characterization of the molecular composition of organic carbon in both gaseous and aerosol is key to understand the processes involved in the formation and aging of secondary organic aerosol. Therefore a technique using active sampling on cartridges and filters and derivatization followed by analysis using a Thermal Desorption-Gas Chromatography/mass spectrometer (TD-GC/MS) has been used to study the molecular composition of organic carbon in both gaseous and aerosol phases during an intensive field campaign which took place in Corsica during the summer 2013: the ChArMEx (Chemistry and Aerosol Mediterranean Experiment) SOP1b (Special Observation Period 1B) campaign.

These measurements led to the identification of 51 oxygenated (carbonyl and or hydroxyl) compounds in the gaseous phase with concentrations comprised between 21 ng m$^{-3}$ and 3900 ng m$^{-3}$ and of 85 compounds in the particulate phase with concentrations comprised between 0.3 and 277 ng m$^{-3}$.



Comparisons of these measurements with collocated data using other techniques have been
conducted showing fair agreement in general for most species except for glyoxal in the gas phase and
malonic, tartaric, malic and succinic acids in the particle phase with disagreements that can reach up
to a factor of 8 and 20 on average, respectively for the latter two acids.
Comparison between the sum of all compounds identified by TD-GC/MS in particle phase with the total
Organic Matter (OM) mass reveal that 18% of the total OM mass can be explained by the compounds
measured by TD-GC/MS for the whole campaign. This number increase to 24% of the total Water
Soluble OM (WSOM) measured by PILS-TOC if we consider only the sum of the soluble compounds
measured by TD-GC/MS. This highlights the non-negligible fraction of the OM mass identified by these
measurements but also the relative important fraction of OM mass remaining unidentified during the
campaign and therefore the complexity of characterizing exhaustively the Organic Aerosol (OA)
molecular chemical composition.
The fraction of OM measured by TD-GC/MS is largely dominated by di-carboxylic acids which
represents 49% of the $PM_{2.5}$ content detected and quantified by this technique. Other contributions to
$PM_{2.5}$ composition measured by TD-GC/MS are then represented by tri-carboxylic acids (15%), alcohols
(13%), aldehydes (10%), di-hydroxy-carboxylic acids (5%), monocarboxylic acids and ketones (3% each)
and hydroxyl-carboxylic acids (2%). These results highlight the importance of poly functionalized
carboxylic acids for OM while the chemical processes responsible for their formation in both phases
remain uncertain. While not measured by TD-GC/MS technique, HUmic-LIke Substances (HULIS)
represent the most abundant identified species in the aerosol, contributing for 59% of the total
identified OM mass on average during the campaign.
14 compounds were detected and quantified in both phases allowing the calculation of experimental
partitioning coefficient for these species. The comparison of these experimental partitioning
coefficients with theoretical ones, estimated by three different models, reveals large discrepancies
varying from 2 to 7 orders of magnitude. These results suggest that the supposed instantaneous
equilibrium being established between gaseous and particulate phases assuming a homogeneous non-
viscous particle phase is questionable.
**1 Introduction**
It is now recognized that aerosols have an impact on human health, climate and ecosystems. However,
large uncertainties still exist on their effects, especially on climate (Fiore et al., 2015). One of the key
solution to reduce these uncertainties is to study the chemical composition of the aerosol organic



fraction since organic aerosols represent a large fraction of fine particles (Jimenez et al., 2009) which
impacts are compound-dependent. Molecular characterization of organic aerosol is therefore crucial.
The OA fraction has been widely studied (e.g. De Gouw and Jimenez, 2009; Fuzzi et al., 2006; Glasius
and Goldstein, 2016; Jacobson et al., 2000; Jimenez et al., 2009; Kanakidou et al., 2005; Pöschl, 2005;
Robinson et al., 2007; Samake et al., 2019; Seinfeld and Pankow, 2003) and many studies allowed to
improve our understanding of their molecular composition (e.g. Gallimore et al., 2017; Nguyen et al.,
2013; Nozière et al., 2015; Zhang et al., 2011b), their sources (e.g. Alves et al., 2012; Jiang et al., 2019;
Shrivastava et al., 2007; Woody et al., 2016), and their formation and evolution processes (e.g. Chacon-
Madrid and Donahue, 2011; Donahue et al., 2012; Heald et al., 2010; Li et al., 2016; Ng et al., 2011).
Organic aerosol can be primary or secondary. Primary Organic Aerosols (POA) are directly emitted in
the atmosphere, whereas Secondary Organic Aerosols (SOA) are formed after oxidation of gaseous
organic precursors such as Volatile Organic Compounds (VOC). These gaseous compounds, coming
from anthropogenic or natural sources, are progressively oxidized by atmospheric oxidants (OH, $O_3$
and $NO_3$). During this multigenerational oxidation process, the O/C ratio of the product formed rises
and their volatility decreases allowing them to condense on existing particles or to form new particles
through nucleation processes (Kulmala et al. 2013), leading to SOA formation. Some of the Semi-
Volatile Organic Compounds (SVOC) formed during the process can be split between the particulate
and gaseous phases. Hamilton et al. (2004) have studied the chemical composition of $PM_{2.5}$ collected
in the urban atmosphere of London using a TD-GCxGC-ToF/MS (Thermal desorption- Gas
ChromatographyxGC-Time of Flight/ Mass Spectrometry) instrument highlighting the presence of
more than 10 000 different organic compounds. In the same study, 130 Oxygenated Volatile Organic
Compounds (OVOC) were also identified while the total number of different VOC in the atmosphere is
estimated to be between 10 000 and 100 000 (Goldstein and Galbally, 2007). The large number of
species composing the gaseous and particulate phases makes an exhaustive characterization of the
atmospheric organic matter challenging.
For this reason, analysis of principal component is often used to describe aerosol composition. This
type of analysis is needed to identify for example the distinction between organic fraction (OC) and
elemental fraction (EC) of carbonaceous particles. Aerosol Mass Spectrometer developed in the early
21$^{st}$ century (Jimenez et al., 2003) allowed making great achievements in our comprehension of aerosol
sources and fate leading to improved agreements between simulated and measured mass
concentrations. This instrument allows measuring the non-refractory fraction of PM, making up the
main components ($NH_4^+$, $NO_3^-$, $SO_4^{2-}$, $Cl^-$, organic matter (OM)) of the fine particles. However, because
of high fragmentation during the ionization step, molecular characterization of the organic fraction is
not possible. Hence, statistical methods were developed to help apportioning the results on organic



matter obtained with this technique. Among them, Positive Matrix Factorization (PMF) applied to
Aerosol Mass Spectrometer (AMS) spectra allows retrieving more information on the sources and
nature of organic aerosol and allows discriminating the total OA into several factors often
characterized by their Oxygen to carbon (O/C) ratio (e.g. Hydrocarbon like Organic Aerosol (HOA), Semi
volatile Oxygenated Organic Aerosol (SV-OOA), Low volatility-OOA (LV-OOA)) (Ng et al., 2010; Zhang
et al., 2011a). Although this classification allows getting insight into the oxidation state of OA, it is not
possible to identify chemical processes involved in SOA formation and aging.
It is therefore essential to perform molecular characterization of organic aerosol. Several techniques
allow this molecular characterization of OA, for example making use of off-line analyses of filter
samplings or online analysis following direct sampling. Coupling Particle Into Liquid Sampler (PILS) to
ion chromatography allow for example the measurement of organic species such as acetate, formate,
oxalate and methane sulfonic acid (MSA) (Orsini et al., 2003; Sciare et al., 2011). Parshintsev et al.
(2009) also coupled PILS with gas chromatography mass spectrometry (GC-MS), which allowed the
measurement of species such as alpha-pinene, pinonaldehyde, cis-pinonic and pinic acids. More
recently, PILS was coupled to ultra-high performance liquid chromatography and electrospray
ionization – quadrupole – time of flight – mass spectrometry (UPLC/ESI-Q-TOF-MS) allowing the
measurement of species as diverse as adenine, adonitol, sorbitol, adipic acid, vanillic acid, azelaic acid
cis-pinonic acid and palmitic acid (Zhang et al., 2016). Several studies also use tandem mass
spectrometry (MS/MS or MS[n]) to get some structural information on compounds present in the organic
aerosol thanks to multiple fragmentation (e.g. Fujiwara et al., 2014; Kitanovski et al., 2011; Liu et al.,
2015; Nguyen et al., 2011). This technique has led to the identification of species such as carboxylic
acids, polycyclic aromatic hydrocarbons (PAH), oxy and nitro-PAH but also oligomers from isoprene
photo-oxidation experiments in the presence of low or high NOx concentrations. Development of
double chromatographic systems (GCxGC or LCxLC) allows reaching lower detection limit separation
capacity and allows measuring a larger range of compounds (Hamilton et al., 2004; Parshintsev and
Hyötyläinen, 2015). Online chromatographic systems also exist to analyze the composition of the
particulate phase. However, difficulties in particle sampling made this type of development
challenging. Williams et al. (2006) developed a thermo-desorption Aerosol GC/MS-Flame Ionization
Detector (FID) allowing the online measurement of compounds of low polarity and with a small
number of chemical functions. GC analysis is usually restricted to compounds of low polarity which
excludes a lot of secondary component of OA. A derivatization step is therefore often used before the
analysis or even during the sampling to perform OA chemical characterization. For example, O-
(2,3,4,5,6-PentaFluoroBenzyl)HydroxylAmine (PFBHA) can be used for measurements of carbonyl
compounds, and N,O-bis(trimethylsylil)-trifluoroacetamide (BSTFA) is used to reduce the polarity of



hydroxyl compounds (Chiappini et al., 2006; Flores and Doskey, 2015; Pietrogrande et al., 2009;
Schoene et al., 1994).
In addition of sample preparation and detection system, different types of extraction systems exist to
avoid multiple steps prior to analysis. For example, Chiappini et al. (2006) have developed a technique
using Supercritical Fluid Extraction (SFE)-GC/MS. With this technique, compounds are extracted from
the filter by supercritical $CO_2$ including a derivatization step with BSTFA as reagent inside the extraction
cell. Extraction efficiency depends on compound solubilities in the supercritical $CO_2$ which has a very
high solvatation power. Thermo-desorption (TD) is another technique allowing to free from
preparation steps prior to analysis. This technique relies on the volatilization of collected compounds
and is suitable for semi-volatile constituent of SOA. It has the advantage to be commercially available
with fully automatized systems, high sensibility allowing the analysis of very low quantity of aerosol
and low preparation time requirement limiting the risk of lost or contamination of analyzed samples
(Hays and Lavrich, 2007; Parshintsev and Hyötyläinen, 2015). This technique has been used by Bates
et al. (2008) and van Drooge et al. (2009) to quantify particulate PAH, while Ding et al. (2009) used it
to measure PAH, alkanes, hopanes and steranes in $PM_{2.5}$.
Although numerous analytical methods exist for SOA chemical characterization, the multiphasic state
of lots of compounds is rarely studied. Indeed, gaseous phase chemical characterization is often
studied separately using techniques such as Proton Transfer Reaction (PTR)/MS (Hansel et al., 1995;
de Gouw and Warneke, 2007; Holzinger et al., 2019) or online/offline GC techniques coupled to various
detectors (e.g. FID, MS) (e.g. Barreira et al., 2015; Kajos et al., 2015; Valach et al., 2014). Despite this
disconnected treatment between aerosol and gaseous phases, understanding mechanisms controlling
the partitioning of SVOC between both phases is key to understand the formation and fate of SOA. A
partition coefficient is defined according to the thermodynamic equilibrium to calculate the mass
transfer of SVOC into particulate phase (Pankow, 1994). This equilibrium is thought to be dominated
by absorption phenomena (Liang et al., 1997) and partition coefficient is therefore calculated
accordingly using models. However, the validity of the instantaneous equilibrium between both phases
as well as the predominance of absorption processes in the mass transfer process are questionable
(Bateman et al., 2015; Fridlind et al., 2000; Healy et al., 2008; Rossignol et al., 2012; Virtanen et al.,
2010). It is therefore crucial to test the theoretical partition coefficient against values measured in the
field for which in situ measurements of organic compounds in both phases are needed.
The Mediterranean Basin is an excellent location to study organic aerosol formation and aging since it
experiences intensive natural and anthropogenic emissions as well as strong photochemistry (Lelieveld
et al., 2002). The ChArMEx project (Chemistry and Aerosols Mediterranean Experiments) aimed at
assessing the present and future state of the atmosphere in the Mediterranean basin. In this frame,





an intensive field campaign was performed at Cape Corsica for 3 weeks during summer 2013 setting
up numerous instruments to investigate the chemical composition of aerosol and gaseous phases.
As part of this project, this study aims at characterizing the molecular composition of organic carbon
in both the gaseous and aerosol phases during the campaign using TD-GC/MS measurements. These
measurements were first compared to measurements performed with other techniques (offline
cartridges analysis using HPLC and GC/FID-MS as well as PTR-MS for gaseous measurements and filter
analysis using Ion chromatography, GC/MS and HPLC). All of these measurements were used to assess
the composition of organic carbon and to estimate the experimental partition coefficient of
compounds measured in both phases to be compared with theoretical values.

## 2 The ChArMEx field campaign

### 2.1 Description of the Cape Corse ground site

The ChArMEx field campaign took place from July 15[th] to August 5[th] 2013 at Ersa in Cape Corsica
(42.97°N, 9.38°E) at the top of a hill (533 meters above sea level). The site is located at the northern
tip of a thin peninsula, a few kilometers from the sea in all directions (between 2.5 and 6 km) and
approximately 30 km north from the nearest urban area (Bastia). Mountains (peaking between 1000
and 1500 m) are limiting transport of urban air masses to the sampling site. The site is surrounded by
typical vegetation of Mediterranean areas (maquis shrubland). Apart from this local biogenic influence,
the site is mainly influenced by marine, and other natural (e.g. dust) emissions, and by continental and
aged air masses due to long range transport. During summer, recirculation of air masses favors
secondary aerosol and ozone build up (Millan et al., 1997). More details about the site, atmospheric
conditions encountered during the campaign and air mass origin can be found in Michoud et al. (2017).

### 2.2 Sampling devices and TD-GC/MS analysis for the molecular characterization of multiphase organic carbon

Simultaneous sampling of gas and particulate phases has been conducted using a parallel sampling
system with two independent pumps allowing the selection of flow rates specifically adjusted for each
phase
Following the sampling, the molecular characterization of gaseous and particulate oxygenated organic
compounds sampled during the campaign has been made using a TD-GC/MS analysis after
derivatization steps following the method developed by Rossignol et al. (2012).

### 2.2.1 Gaseous phase

2.2.1.1 Gaseous phase sampling





Sampling of gaseous oxygenated compounds was achieved by using commercial sorbent cartridges
containing Tenax TA (porous polymers based on 2.6-diphenyl-p-phenylene oxide; Perkin Elmer™ or
Markes™) that has been previously impregnated with suitable derivatization agents (see below)
following an improved protocol from Rossignol et al. (2012). To maximize the adsorption surface, small
particle size of 60/80 mesh has been selected. Ambient air samplings were performed during 6h at a
flow rate of 100 mL min$^{-1}$. A Teflon filter (Zefluor™ membrane, Pallflex™, 47 mm) was installed
upstream from the cartridges to trap particulate compound that could potentially been adsorbed on
the Tenax adsorbent. Gaseous phase sampling has been performed using individual pumps (Gilian™
pump, model LFS-113DC). Prior to sampling, cartridges were heated at 320°C under a small helium
flow rate during 4h to eliminate any trace of contamination. Every single cartridge was then analyzed
to ensure its cleanliness with quantities below Limit of Detection (LOD) for all measured compounds.
During the campaign, 177 gaseous samples were collected following this protocol.
2.2.1.2 Sample preparation for gaseous phase
For the analysis of multi-functionalized OVOC by gas chromatography, a derivatization step is needed.
It allows the suppression of the reactivity of functions, improving their thermal stability and rising their
volatility. The dual derivatization reagents used in this study are PFBHA for carbonyl compounds and
MTBSTFA (N-tert-Butyldimethylsilyl-N-methyltrifluoroacetamide) for hydroxyl compounds. The two
derivatization processes are performed separately.
2.2.1.2.1 Carbonyl compounds
PFBHA has been used as derivatization reagent for the analysis of carbonyls. Cartridges have been
impregnated prior to sampling thanks to a glass balloon with 8 arms, containing 0.33mg of solid PFBHA
per cartridges mounted on the balloon, and on which the cartridges are installed under a 100 mL min$^{-1}$
nitrogen flow rate per cartridges at 110°C during 20 minutes. The impregnated cartridges are stored
at room temperature until the sampling. After sampling, cartridges are stored at room temperature
during 5 days, optimum for the derivatization step using PFBHA (Ho and Yu, 2002), before their
analysis.
2.2.1.2.2 Hydroxyl compounds and carboxylic acids
MTBSTFA with 1% of TBDMCS (tert-butyldimethylchlorosilane, used as catalyst for the reaction) has
been used as derivatization agent for the analysis of hydroxyl compounds. Cartridges are impregnated
prior to sampling vaporizing 0.3 μL of MTBSTFA at 275°C using a commercial thermal tube desorber
(Dynatherm Analytical Instruments, model 890) under a flow of Helium of 30 mL min$^{-1}$ for 11 minutes.
The cartridges are then stored at room temperature and sampling is performed within 10 days after
impregnation. After sampling, cartridges are stored at 4°C. To ensure complete derivatization of all



compounds before the analysis, two deposits of 0.3 µL of MTBSTFA are achieved on each side of the
cartridges which are kept at 60°C during 5h after that. Once the cartridges are back at room
temperature, analysis is performed within 5 hours.
**2.2.2 Particulate phase**
2.2.2.1 Particulate phase sampling
Sampling of particulate matter was performed over regular (not impregnated) filters and derivatization
was performed only after sampling (to avoid chemisorption of gaseous compounds on filters) following
a protocol adapted from Rossignol et al. (2012). The sampling device used during the campaign was a
modified Speciation Sampler Partisol, model 2300 (Rupprecht & Patashnick Co, Thermo Fisher
Scientific). Three ChemComb cartridges, with $PM_{2.5}$ impactors, were mounted to this device to allow
the sampling of particulate phase on filters of different nature according to targeted compounds. For
carbonyls compounds and non-oxygenated compounds Quartz filters (Pallflex™, 47 mm) were used.
For hydroxyl compounds, quartz filters are not suitable because of silanol groups present at their
surfaces that can be derivatized instead of the hydroxyl compounds reducing considerably their
derivatization yield (Rossignol, 2012). Therefore, for the sampling of this type of compounds, we
selected filters of borosilicate glass fibers coated with tetrafluoroethylene (TFE) called hereafter
"Teflon quartz filters" (Fiber film, Pallflex™, 47mm). Activated carbon honeycomb denuders were
installed upstream from the filters to avoid positive artifacts due to adsorption of gaseous oxygenated
compounds on the filters. For cleaning and a best efficiency, denuders were heated at 250°C before
being used for each new sample. The sampling flow rate was of 1 $m^3$ $h^{-1}$ for each sample step. Quartz
and Teflon quartz filters were carbonized prior to the sampling respectively at 500°C and 300°C to
eliminate any possible contamination. During the campaign, 240 particulate samples were collected
following this protocol.
2.2.2.2 Sample preparation for particulate phase
2.2.2.2.1 Carbonyl compounds
Sampling are performed on quartz filters which are stored at -16°C after sampling waiting for analysis.
Then, the filters are cut into two pieces, both inserted into empty and clean stainless steel tubes. These
tubes, including grids, are previously sonicated in several bath of ultra-pure water and acetonitrile and
then are heated at 400°C under a flow of helium (80mL $min^{-1}$) during 4h. Deposition of 50 µL of PFBHA
saturated solution (acetonitrile/water (90/10, v/v) with 27 mg $mL^{-1}$ of PFBHA) are achieved in the tubes
to expose adsorbed compounds to the derivatization reagent. Tubes are then stored at room
temperature during 5 days to allow derivatization of adsorbed compounds before their analysis.
2.2.2.2.2 Hydroxyl compounds and carboxylic acids



Sampling are performed on Teflon quartz filters which are stored at -16°C after sampling waiting for
analysis. Derivatization is performed after sampling directly on filters. Filters are put in stainless steel
tubes cleaned following the same protocol than for carbonyl compounds. Tubes are then sealed and
maintained vertically with 10 µl of MTBSTFA put in the bottom cap for passive impregnation during
24h at room temperature.
**2.3.3 Analytical system**
The analytical system used in this study is composed by three successive modules: a thermal
desorption system, a gas chromatography unit and a mass spectrometer.
The thermal desorption allows the extraction of adsorbed compounds on sample support by increasing
the temperature without any preliminary solvent extraction and collecting them on a cold trap before
flash injection in GC/MS instrument. The thermal desorption system (Markes™, model unity 1) is
coupled with an automated system (Markes™, model Ultra 50:50). Thermal desorption parameters are
listed in Table 1.
The GC/ MS instrument (Agilent Technologies Inc.) used during this study is composed by two modules:
- A GC unit, model 6890 A, associated with a capillary column Integra-Guard Rxi®-5Sil MS
(stationary phase: 1.4-bis(dimethylsiloxy)phenylene dimethyl polysiloxane, length: 60m,
diameter: 250µm, film thickness: 0.25µm, with 5m pre-column deactivated without any
stationary phase; Restek Corporation).
- A Mass spectrometer, model 5973N, equipped with an ionization source in EI (Electronic
Impact) or CI (Chemical Ionization; using $CH_4$ as reagent gas) and associated with a quadrupole.
GC/MS parameters are listed in Table 1.
**2.3.4 Internal calibration protocol**
For a more efficient quantification, internal calibration has been set up for both family of compounds
(carbonyl and hydroxyl) and for both phases. This procedure aims at taking into account drift in MS
sensitivity and derivatization efficiency. Two types of internal standards are used: substitutes which
are deuterated compounds getting at least one derivatized function; and an internal standard which is
a compound with no derivatized function. 50 ng of Substitutes are added prior to the derivatization
step to take into account every steps of sample preparation as well as analysis steps. The list of
substitutes selected is given in Table 2. The internal standard selected is pentadecane and 50 ng is
added on cartridges grid just before the analysis.
**2.3.5 Estimation of uncertainties**



Overall uncertainties have been determined taking into account precision, detection limit and
systematic errors (including uncertainties on standard concentrations, on calibration, on blank
determination and on sampling volume; following Gaussian error propagation). Overall uncertainties
have therefore been estimated to be 35% and 54% on averaged in gas phase for carbonyls and
hydroxyls and carboxylic acids respectively and to be 41% and 47% on averaged in particulate phase
for carbonyls and hydroxyls and carboxylic acids respectively.
**2.4 Ancillary measurements**
An important set of complementary instruments, dedicated to the measurement of both gaseous and
particulate phase, has been deployed at the supersite supporting the interpretation and validation of
the TD-GC/MS dataset.
**2.4.1 Gaseous ancillary measurements**
2.4.1.1 PTR-MS
Measurements of OVOCs (e.g. nopinone, sum of methacrolein and methyl vinyl ketone, propanoic acid
and methyl ethyl ketone), among other species (e.g. aromatics and biogenic VOCs) were performed
using a Proton Transfer Reaction-Time of Flight Mass Spectrometer (PTR-ToF-MS, KORE Inc® 2nd
generation). A detailed description of these measurements was given by Michoud et al. (2017, 2018).
Briefly, ambient air was sampled through a 5-m long Teflon PFA (PerFluroAlkoxy) line held at 50°C at a
flow rate of 1.2 L min$^{-1}$, leading to a residence time of 3.1s in the sampling line. The PTR-ToF-MS
sampling flow rate was set at 150 mL min$^{-1}$. The instrument was operated at a reactor pressure and a
temperature of 1.33 mbar and 40°C, respectively, leading to an E/N ratio of 135 Td.
An automated zero procedure was performed every hour for 10 min. Humid zero air was generated by
passing ambient air through a catalytic converter to perform zeros at the same relative humidity than
ambient air.
Signals from protonated VOCs were normalized by the signals of $H_3O^+$ and the first water cluster
$H_3O^+(H_2O)$ as proposed by de Gouw and Warneke (2007). Concentrations were calculated using Eq. (1):

$$[R] = \frac{i_{R\_net}}{(i_{H_3O^+} + X_r.i_{H_3O^+(H_2O)})} \cdot \frac{150000}{R_{f,R}} \tag{1}$$

Where [R] represents the mixing ratio of a given VOC, $i_{R\_net}$ the net signal of this VOC, $i_{H3O+}$ and $i_{H3O+(H2O)}$
the signals of $H_3O^+$ and $H_3O^+(H_2O)$. Xr is a factor introduced to account for the effect of humidity on the
PTR-MS sensitivity (de Gouw and Warneke, 2007) and is determined experimentally through
calibrations performed at various relative humidity. $R_{f,R}$ is the sensitivity determined during calibration
experiments (in ncts ppt$^{-1}$) and normalized to 150 000 counts s$^{-1}$ of $H_3O^+$ ions. The latter is the number



of counts of reagent ions (not corrected for ion transmission into the ToFMS) observed on this PTR-
ToF-MS instrument. Data were recorded at a time resolution of 1 min. During the campaign,
calibrations were performed every 3 days using various standards, including a canister containing 15
VOCs (NMHCs, OVOCs and chlorinated VOCs; Restek®), a gas cylinder containing 9 NMHCs (Praxair®)
and a gas cylinder containing 9 OVOCs (Praxair®). Information about the composition of these
standards can be found in Michoud et al. (2017). Overall uncertainties are estimated between 6 and
23% depending on the compound considered (Michoud et al., 2017) following the "Aerosols, Clouds,
and Trace gases Research InfraStructure network" (ACTRIS) guidelines for uncertainty evaluation
(ACTRIS, 2012).
2.4.1.2 GC-FID/MS
OVOCs, including aldehydes, ketones, alcohols, ethers, esters, as well as a few NMHCs, including BVOCs
and aromatics, were measured using an online GC/FID-MS instrument. This instrument as well as its
setup during the campaign was described by Michoud et al. (2017). Briefly, ambient air was sampled
via a KI ozone scrubber and a 5-m long PFA line (1/8") at a flow rate of 15 mL min$^{-1}$ using an Air server-
unity I (Markes International®). The sample was pre-diluted (50% dilution) with dry zero air to keep
relative humidity below 50%. The sample was then collected in an internal trap, consisting in a 1.9 mm
i.d. quartz tube filled with two different sorbents (5 mg of Carbopack B and 75 mg of Carbopack X,
Supelco®) and cooled at 12.5 °C by a Peltier system. Compounds trapped on the sorbents were then
thermally desorbed at 280 °C and injected into the column of a GC (Agilent®) equipped with a FID for
detection and quantification and with a Mass Spectrometer (MS) for identification. The compounds
were separated through a high polar CP-lowox column (30 m×0.53 mm× 10 μm) (Varian®). The time
resolution of these measurements is 1h30min. Calibrations were performed during the campaign using
a gas cylinder containing 29 VOCs (Praxair). Information about the composition of this standard can be
found in Michoud et al. (2017). Overall uncertainties are estimated between 5 and 14% depending on
the compound considered (Michoud et al., 2017) following ACTRIS guidelines for uncertainty
evaluation (ACTRIS, 2012).
2.4.1.3 Active sampling on DNPH cartridges
Carbonyl compounds were collected continuously for 3 h durations by active sampling on DNPH
cartridges (Waters®) using an automatic sampler (Tera Environment®). These compounds were
analyzed later by High Performance Liquid Chromatography (HPLC) with UV detection. Ambient air was
sampled via a 3-m PFA line (1/4") at 1.5 L min$^{-1}$ and passed through a KI ozone scrubber and a stainless-
steel particle filter (porosity: 2μm). More details about these measurements are given by Michoud et
al. (2017; 2018). Calibrations were performed at the laboratory using Supelco® standard for DNPH.



Overall uncertainties are estimated around 25% (Michoud et al., 2017) following ACTRIS guidelines for
uncertainty evaluation (ACTRIS, 2012).
2.4.1.4 Inorganic trace gases
During the campaign, NO and $NO_2$ were measured by a commercial ozone chemiluminescence analyzer
(Cranox II; Eco Physics®) with a time resolution of 5 min. NO was measured directly, while $NO_2$ was
converted into NO using a photolytic converter. $O_3$ was measured using a commercial analyzer (TEI 49i;
Thermo Environmental Instruments Inc®) using UV absorption with a time resolution of 5 min.
**2.4.2 Particulate ancillary measurements**
Mass concentrations of $PM_{10}$ and $PM_1$ were measured during the campaign using two tapered element
oscillating microbalance (TEOM) equipped with a filter dynamic measurement system (FDMS) (Thermo
Scientific™). In addition, aerosol chemical composition was measured by online technique (aerosol
chemical speciation monitor - ACSM) and offline-method (Ion chromatography, GC/MS and HPLC) on
filters collected daily with 2 HiVol samplers (30 $m^3$ $hr^{-1}$) equipped with $PM_1$ and $PM_{2,5}$ inlets.
2.4.2.2 ACSM
Measurements of the chemical composition of non-refractory submicron aerosol (NR-$PM_1$) have been
carried out using a quadrupole ACSM (Aerodyne Research Inc., Billerica, MA, USA). These
measurements have been described in detail by Michoud et al. (2017). Briefly, the calibration of this
instrument with monodispersed (300 nm diameter) ammonium nitrate particles was performed 2
months before the campaign. Because ambient air was dried by a Nafion membrane and because
ammonium nitrate was low during the campaign, constant collection efficiency (CE) of 0.5 has been
kept. The Q-ACSM was operated continuously during the whole campaign at a time resolution of 30
min.
2.4.2.3 Ion Chromatography
Soluble anions and cations were analyzed by ionic chromatography (IC, ThermoFisher ICS3000)
following protocol similar to that described elsewhere (e.g. Jaffrezo et al., 1998). Briefly, 38 mm
diameter sub-samples from each filter were soaked for 20 min in 10 mL of Milli-Q water with orbital
shaking, and then filtered using 0,22 µm-porosity Acrodisc filters before analysis. ASA11-HC and CS16
columns were used for anions and cations analyses, respectively.
2.4.2.4 GC/MS
Organic markers were analyzed by gas chromatography (GC) coupled with mass spectrometry (MS)
using the method developed by El Haddad et al. (2011). Filter samples were first spiked with 300µL of
a solution containing the internal standard D6-Cholesterol ($C_{24}H_{40}D_{60}$). Accelerated Solvent Extraction


(ASE Dionex 300) was performed with a mixture of acetone/dichloromethane (1/1 v/v) at 100bar and
100°C during 10 min. Sample extracts were concentrated using a Turbo Vap II under $N_2$ in a water-bath
regulated at 40°C to a final volume of 500µL. A fraction of the extracts (50µL) was derivatized at 70°C
for 90 min by adding 100µL of N,O-bis(triméthylsilyl)trifluoroacétamide (BSTFA containing 1% of
TMCS). Derivatized extracts were then analyzed using a Thermo Trace Ultra GC coupled with a Polaris
Q – ion trap operating in the electron impact mode. The GC was equipped with a TR-5MS capillary
column (30 m × 0.25 mm i.d. × 0.25 µm film thickness). Aliquots of 1 µL were injected in split mode
(split ratio 50) at 280°C. The column temperature program was held at 65°C hold for 2 min, and ramped
at 6°C/min up to 300°C, followed by an isothermal hold at 300°C for 20 min. GC-MS response factors
were determined using authentic standards. Compounds, for which no authentic standard are
available, were quantified using the response factor of compounds with analogous chemical
structures. Field blank filters were also treated with the same procedure.
2.4.2.5 HPLC
The analysis of a large array of organic acids (including pinic and phthalic acids, and 3-MBTCA) was
conducted using the same water extracts as for IC and HPLC-PAD analyses. In brief, this was performed
by HPLC-MS (GP40 Dionex with a LCQ-FLEET Thermos-Fisher ion trap), with negative mode
electrospray ionization. The separation column is a Synergi 4 µm Fusion – RP 80A (250×3 mm ID, 4 µm
particle size, from Phenomenex). An elution gradient was optimized for the separation of the
compounds, with a binary solvent gradient consisting of 0.1% formic acid in acetonitrile (solvent A)
and 0.1% aqueous formic acid (solvent B) in various proportions during the 40-minute analytical run.
Column temperature was maintained to 30 °C. Eluent flow rate was 0.5 ml min⁻¹, and injection volume
was 250 µl. Calibrations were performed for each analytical batch with solutions of authentic
standards. All standards and samples were spiked with internal standards (phthalic-3,4,5,6-d4 acid and
succinic-2,2,3,3-d4 acid). The calculation of the final atmospheric concentrations was corrected with
the concentrations of internal standards and of the procedural blanks, taking also into account the
extraction efficiency varying between 76-116% (depending on the acid).
2.4.2.6 OCEC SUNSET field instrument
Concentrations of elemental carbon (EC) and organic carbon (OC) in $PM_{2.5}$ were obtained in the field
from an OCEC Sunset field instrument (Sunset Laboratory, Forest Grove, OR, USA; Bae et al., 2004)
operated at a flow rate of 8 L min⁻¹ with a denuder set upstream to avoid adsorption of semi-volatile
compounds on the filter collecting particles in the instrument. Data were obtained every 2 hours with
this instrument.
2.4.2.7 PILS-TOC


PM$_1$ water-soluble organic compounds (WSOCs) were measured by a modified PILS (Brechtel
Manufacturing Inc., USA; Sorooshian et al., 2006) coupled with an analyzer of total organic carbon
(TOC; model Sievers 900; Ionics Ltd, USA). Sciare et al. (2011) and Michoud et al. (2017) described this
technique and operating procedures used during the ChArMEx field campaign. Briefly, the PILS-TOC
instrument was operated at a flow rate of 15 L min$^{-1}$ with a dilution factor of 1.30. A 0.45 μm pore size
diameter filter in polyethylene was set in-line in the aerosol liquid flow to analyze the water-soluble
OC fraction only and a VOC denuder was set upstream the collection to avoid semi-volatile VOC
contamination. Daily blanks were conducted every day for 1h by placing a total filter upstream of the
sampling system.
2.4.2.8 HULIS measurements
The water soluble HULIS fraction is analyzed according to a protocol described in detail in Baduel et al.
(2009). Briefly, the water-soluble fractions obtained from aerosol samples are passed through a weak
anion exchange resin (GE Healthcare®, HiTrap™ DEAE FF, 0.7cm ID x 2.5cm length) without any pre-
treatment. After this concentration step, the organic matter adsorbed is washed with 12mL of a
solution of NaOH 0.04M (J.T.Baker®, pro analysis) to remove neutral components, hydrophobic bases,
inorganic anion, mono- and di-acids initially retained in the resin. Finally, HULIS$_{WS}$ are quickly eluted in
a single broad peak using 4 mL of a high ionic strength solution of NaCl 1M (Normapur®). All flow rates
are set at 1.0 mL min$^{-1}$. UV-Vis absorption spectra are measured on-line after the extraction system,
using a diode array detector (Dionex UV-VIS 340U), and recorded in the range 220-550nm. The HULIS$_{WS}$
fraction is subsequently collected manually and the carbon content is analyzed with a DOC analyser
(Shimadzu TOC-V$_{CPH/CPN}$) by catalytic burning at 680°C in oxygen followed by non-dispersive infrared
detection of the evolved CO$_2$.
**3 Results and discussion**
**3.1 Main conditions during the campaign**
**3.1.1 Meteorological conditions**
Meteorological and environmental conditions are presented in Table 3. Relatively high temperatures
were monitored during the campaign (up to 32°C) coinciding with high biogenic emissions from local
vegetation and strong photochemistry (Michoud et al., 2017). These conditions led to high ozone
concentrations during the campaign (65 pbbv on average for the overall sampling period and up to 111
ppbv for 5 min measurements), typical of this region during summer (e.g. Lelieveld, 2002; Di Biagio et
al., 2015). High relative humidity was encountered at night with values reaching 100% coinciding with
foggy conditions observed during several nights at the site. High wind speeds were monitored with
maximum reached on the 30$^{th}$ of July 2013 (13.2 m s$^{-1}$). During the campaign, almost 40% of air masses



came from the south-west sector and 20% from the western sector (see Figure 1). Winds coming from
south-west sector are predominant during daytime and nighttime and correspond to wind speed
maxima. Winds from the west and north-east are also recorded, but during daytime only. Low $NO_x$
concentrations were observed during the campaign (0.57 ppbv on average) with a few spikes above 1
ppbv corresponding to local influence from traffic especially when air masses came from the south
(e.g. 27[th] July).

### 3.1.2 Particles and organic fraction

Mean, median, maximum and minimum of mass concentrations of $PM_{10}$, $PM_1$ and organic fraction in
NR-$PM_1$ are summarized in Table 3 for the whole campaign. The averaged mass concentrations for
$PM_{10}$ is 12.0 µg m$^{-3}$, comparable to observations performed at other remote sites located in the
western Mediterranean basin (e.g. 15.5 µg m$^{-3}$ at Montseny, Spain; 11.5 µg m$^{-3}$ between 2010 and
2013 at Montsec, Spain; 14.6 µg m$^{-3}$ at Monte Martano, Italy; 13 µg m$^{-3}$ between 2010 and 2013 at
Venaco, France – Moroni et al., 2015; Nicolas, 2013; Querol et al., 2009a, 2009b; Ripoll et al., 2015).
The averaged mass concentrations for $PM_1$ was 8.3 µg m$^{-3}$ during the campaign and represented an
important fraction of $PM_{10}$ (69% on average). The amount of $PM_1$ at Ersa is also comparable to what
has been previously measured in other remote sites in the western Mediterranean basin (e.g. 8.2 µg
m$^{-3}$ at Montseny, Spain; 7.1 µg m$^{-3}$ between 2010 and 2013 at Montsec, Spain – Minguillón et al., 2015;
Ripoll et al., 2015). During the campaign, the organic fraction represented between 40 and 55% of $PM_1$
mass concentrations (mean of 3.7 µg m$^{-3}$ representing 44% of $PM_1$ on average).
Time series of mass concentrations of $PM_{10}$, $PM_1$ and organic fraction in $PM_1$ are presented in Figure
2. Highest mass concentrations for $PM_{10}$ and $PM_1$ are observed between 12 and 21 July (15.7 and
11.0 µg m$^{-3}$ on average respectively for $PM_{10}$ and $PM_1$). According to back trajectory analysis (Michoud
et al., 2017) this period corresponds to low wind speed and hence stationary air masses. A decrease of
$PM_{10}$ concentrations is observed from 21 to 25 July (12.0 µg m$^{-3}$ on average) while the ratio $PM_1/PM_{10}$
and organic/$PM_1$ are the highest. During this period, the $PM_{10}$ and $PM_1$ fractions are almost the same.
This period is characterized by higher wind speed and air masses coming from the north-eastern sector
and therefore characterized by anthropogenic influence from northern Italy. From 26 to 29 July, a rise
in $PM_{10}$ mass concentrations is observed coinciding with the warmest temperature of the campaign
and air masses coming from the south and characterized by biogenic influence (see Michoud et al.,
2017). From 29 July to 3 August, $PM_1$ concentrations strongly decrease (from 9.3 to 2.6 µg m$^{-3}$ on
average) coinciding with higher wind speed and relative humidity while winds came from north-west
and north-east directions (see Michoud et al. 2017). During the last period (3-5 August), increase of
$PM_{10}$ and $PM_1$ concentrations is observed and a clear diurnal cycle is monitored for both fractions





corresponding to a raise in temperatures. Overall, the organic fraction evolution follows the one of the
$PM_1$ mass fraction.
**3.2 Results from the TD-GC/MS analysis**
**3.2.1 Compound identifications**
Detection of functionalized compounds led to the identification of 23 carbonyl compounds and 28
hydroxyl compounds and carboxylic acids in the gaseous phase and of 30 carbonyl compounds and 55
hydroxyl compounds and carboxylic acids in the particulate phase. The entire list of these 97
compounds is presented in supplementary material 1 together with their O/C ratio, their calculated
saturation vapor pressure, the main fragments of their mass spectra, the method used for their
identification, the substitute used to account for the derivatization efficiency, the external standard
used for their quantification, the fragment used for quantification and the averaged concentrations
measured in both phases. For the carbonyl compounds, the mono-functionalized compounds
identified contained from 3 (e.g. propanal) to 10 (e.g. decanal) carbon atoms and from 2 (e.g. glyoxal)
to 5 (e.g. 4-oxopentanal) carbon atoms for the bi-functionalized compounds. For the hydroxyl
compounds and the carboxylic acids, the mono-functionalized identified compounds contained from
3 (e.g. propanoic acid) to 18 (e.g. octadecanoic acid) carbon atoms. Several poly-functionalized
compounds have also been identified: hydroxy-acids and di-acids from 2 (e.g. glycolic acid) to 8 (e.g.
mandelic acid) carbon atoms; triols, di-hydroxy-acids, hydroxyl-di-acids, tri-acids from 3 (e.g. glycerol)
to 9 (e.g. 2-Hydroxy-4-isopropyl-hexanedioic acid) carbon atoms; and two tetra-functionalized
compounds (methyl-tetrols and citric acid).
It is worth noting that several compounds exhibited very close quantities in the air sample and in the
blank (designed as "blank" in the supplementary material 1). Therefore, the presence of these
compounds in the air sampled cannot be certain. For the compounds that have been quantified
successfully and present concentrations significantly above the quantification limit ($3\sigma$ above averaged
blank measurements), higher levels are observed in the gas phase. The averaged concentrations
ranged from 21 ng m$^{-3}$ (Mandelic acid) to 1600 ng m$^{-3}$ (glycerol) for hydroxyl compounds in the gas
phase and from 0.3 (Pyruvic acid) to 277 (oxalic acid) ng m$^{-3}$ in the particulate phase. For the carbonyl
compounds, the averaged concentrations ranged from 85 ng m$^{-3}$ (hexanone) to 3900 ng m$^{-3}$ (4-
Oxopentanal) in the gas phase and from 1 ng m$^{-3}$ (e.g. methylpropanal or glyoxal) to 20 ng m$^{-3}$ (4-
methylpentanal) in the particulate phase. Figure 3 presents the distribution of all quantified
compounds along their saturation vapor pressure and their O/C ratio. The phases in which these
compounds were identified are also shown in Figure 3. While compounds only present in the gas or
aerosol phase exhibit high and low saturation vapor pressure, respectively, some exceptions are
noticeable. Indeed, some gaseous compounds have low vapor pressure (down to $10^{-8.6}$ atm) such as



long chain linear mono carboxylic acids (up to 15 carbon atoms) and some compounds only found in
the particle phase have high vapor pressure (up to $10^{-0.8}$ atm), normally incompatible with their
presence in such phase, such as small mono carbonyls (e.g. methylpropanal, methylbutanone, 2-
methylbutanal…). We also found compounds exhibiting high vapor pressure (up to $10^{-0.4}$ atm) in both
phases, which is normally incompatible with their presence in aerosol phase, such as small carbonyls
(e.g. propanal, acrolein, methacrolein, MVK…). This latest point is discussed further in section 3.2.5.
**3.2.2 Data intercomparison**
A comparison of data measured by TD-GC/MS with other techniques available on site has been
performed, for both phases, to test the reliability of these measurements.
3.2.2.1 Gas phase
Comparisons of TD-GC/MS data with PTR-ToF-MS and GC/FID/MS data averaged over the same
sampling duration at a similar time step have been performed and are shown in Figure 4 and Figure 5.
Fair agreement is found for nopinone, the sum of methacrolein and methyl vinyl ketone, propanoic
acid and methyl ethyl ketone between TD-GC/MS measurements and measurements performed by
PTR-ToF-MS. Good agreement is also found for methyl vinyl ketone and 2-hexanone between TD-
GC/MS measurements and measurements performed by GC/FID/MS. Ranges of measured
concentrations are similar between these techniques as well as the temporal variation.
Comparisons of TD-GC/MS measurements with DNPH cartridges analysis are presented in Figure 6. For
these latter, only the first ten days of the campaign have been validated because of a leak issue in the
sampling system of DNPH cartridges after that period (see Michoud et al., 2017). Ranges of
concentrations are in the same order of magnitude between these two techniques for propanal,
acrolein, methacrolein, methyl ethyl ketone, methylglyoxal, hexanal and benzaldehyde even though it
is difficult to conclude on their co-variation regarding the small number of data available and the low
time resolution for these two techniques. However, glyoxal and methyl vinyl ketone present large
differences between the two techniques (factor of 15 and 12 respectively). For glyoxal, Matsunaga
(2004) recorded maximum concentrations of 154 ng m$^{-3}$ (≈65 pptv) at a forested site at Moshiri in
Hokkaido island, in summer. Washenfelder et al. (2011) recorded maximum glyoxal concentrations of
500 pptv at an urban site in Los Angeles in summer, while numerous glyoxal precursors exist in urban
environment. Therefore, the concentrations measured by TD-GC/MS seem overestimated and
measurements from DNPH cartridges analysis seem more consistent with these previous observations.
Thermo-degradation of other heavier compounds adsorbed on the Tenax cartridges leading to glyoxal
could be an hypothesis for this overestimation. In the case of methyl vinyl ketone, the good agreement
observed between TD-GC/MS measurements and GC/FID/MS ones (see Figure 5) tends to indicate that



the disagreement observed here is related to an underestimation of the concentrations measured by
DNPH cartridge analysis. Furthermore, recent studies on humidity dependence of the DNPH–HPLC–UV
method for some ketone compounds, revealed that the collection efficiency is inversely related to
relative humidity, with up to 35 %–80 % of the ketones being lost for RH values higher than 50 % at
22 °C (Ho et al., 2014). Furthermore, dimerization issues for MVK during analyses using DNPH method
has also been identified, during more recent measurements, that can cause strong underestimation of
this technique (>50%).
3.2.2.2 Particulate phase
Comparisons of results from filter analysis by TD-GC/MS and by Ion chromatography, GC/MS and HPLC
have been performed and are shown in Figure 7 and Figure 8. The range of concentrations between
TD-GC/MS analysis and other techniques are in the same order of magnitude for oxalic acid, pinic acid,
2-methylglyceric acid, MBTCA, glycolic acid and phtalic acid. However, a discrepancy is found for
malonic acid and tartaric acid which measurements differ both of a factor of 4 on averaged between
TD-GC/MS and HPLC analyses. For methyl-tetrols, the analysis performed by TD-GC/MS did not allow
to distinguish the two isomers. Temporal evolution of compounds shown in Figure 7 and Figure 8 are
also similar from one technique to another, especially for oxalic acid and pinic acid.
Nevertheless, larger disagreements have been observed for some compounds (see Figure 8). An
overestimation of TD-GC/MS analysis compared to HPLC analysis of a factor of 8 and 20 on average,
respectively for malic acid ad succinic acid, is observed. For malic acid, the external standard used for
the estimation of the response factor (glycolic acid) is maybe not appropriate which may explain this
discrepancy. As a test, succinic acid and glutaric acid (two other di-acids) have been used as external
standard for malic acid quantification with no improvement in the agreement observed. For succinic
acid, the authentic standard has been used and such problem cannot explain the discrepancy
observed. No interference in the peak region is observed and this cannot neither explain the
differences observed.
On the whole, comparisons of TD-GC/MS with other techniques deployed during the campaign are
satisfactory for both phases with results at least in the same order of magnitude for the measured
absolute concentrations, except for some compounds. Therefore, these observations allow us to use
TD-GC/MS data both in gas and aerosol phase to study further the behavior of organic carbon at a
molecular level at cape Corsica during ChArMEx campaign, keeping however in mind the potential
biases revealed during this data comparison exercise.
**3.2.3 Description of organic compounds behaviour during the campaign**





Time series of every compounds measured by TD-GC/MS in both phases are presented in the
supplementary material 2.
Concerning the gaseous phase, several linear mono-aldehydes ($C_3$ to $C_{10}$) have been detected and
quantified in the same range of concentrations than what has been previously reported by the same
technique at another site in Corsica (Rossignol et al., 2016). These compounds are mainly primary
compounds emitted by vegetation under stress conditions. For propanal and butanal, some chemical
processes and anthropogenic primary sources (especially ship emission) can also be involved (Agrawal
et al., 2008). During the campaign, these compounds are characterized by daily maxima during daytime
and daily minima during nighttime, confirming the predominance of biogenic sources. This diurnal
cycle is also found when these compounds are also measured in the particulate phase, which may
indicate a thermodynamic equilibrium for these compounds between both phases. Their
concentrations are higher at the end of the campaign (30th of July) coinciding with the warmest period
suggesting higher local biogenic emission.
At the end of the campaign, an elevation of concentrations is also observed for nopinone, 4-
oxopentanal, 2-propenoic acid, methacrylic acid, mandelic acid, glycolic acid and levulinic acid (see
supplementary material 2), all known as oxidation products of biogenic compounds. For example,
Nopinone is an oxidation product of beta-pinene and 4-oxopentanal is known to be an oxidation
product of several biogenic compounds such as squalene and limonene (Fruekilde et al., 1998;
Matsunaga et al., 2004; Rossignol et al., 2012). During this period, air masses were coming from the
southern sector and travelled during a short period of time (12 to 24h) above Corsica and Sardinia
(Michoud et al., 2017; Zannoni et al., 2017). An increase of concentrations is also observed for some
monocarboxylic acids such as propanoic acid, pentanoic acid, hexanoic acid, tridecanoic acid,
tetradecanoic acid and pentadecanoic acid (see supplementary material 2). Several sources are
possible for these compounds that can be either primary or secondary and either biogenic or
anthropogenic, especially for small carboxylic acids ($C_3$ to $C_6$; Chebbi and Carlier, 1996). Longer chain
carboxylic acids are often considered as primary compounds both from biogenic and anthropogenic
sources. Nevertheless, the results we obtained here underline the ubiquitous nature of organic acids
(including long chains) in the atmosphere. It is remarkable to observe that despite their widespread
detection, the knowledge on their sources (including chemical processes) remain scarce. Ozonolysis of
alkenes, reactions between aldehydes and $HO_2$, or hydrolysis of oligomers could be involved.
At the beginning of the campaign (from 13th to 15th July) we observed a rise in concentrations of 4-
oxopentanal, 2-hexanone, glycolic acid, 2-propenoic acid and monocarboxylic acids from $C_3$ to $C_7$ ((see
supplementary material 2). A spike of methacrolein is also observed the 13th of July, highlighting local
emission of biogenic precursors as it is during the calm low wind cluster period (Michoud et al, 2017).



Concerning particulate compounds, observations are different than for gaseous compounds. Indeed,
an important peak of concentrations is observed for many compounds from 17th to 19th of July, e.g. 3-
isopropylglutaric acid, 3-hydroxy-4,4-dimethylglutaric acid, ketonorlimonic acid, ketolimonic acid,
tricarballylic acid and methyltartronic acid (see supplementary material 2). The four first compounds
correspond to oxidation products of biogenic precursors such as pinenes and limonene. O/C ratios for
these compounds are high, varying from 0.5 (3-isopropylglutaric acid) to 1.3 (methyltartronic acid).
This period corresponds to a rise in aerosol mass concentration (see Figure 2), with stagnant air masses
and very low wind speed (Michoud et al., 2017). Associated with strong photochemistry, this favored
chemical processing and the formation of secondary products with high O/C ratio. Other compounds
also show a rise in their concentrations at this time (see supplementary material 2): unsaturated
carboxylic acids (crotonic acid, 2-hydroxy-3methyl-2-pentenoic acid), long-chain monocarboxylic acids
(hexadecanoic acid and octadecanoic acid), dicarboxylic acids (malonic acid, succinic acid, glutaric
acid), unsaturated dicarboxylic acids (maleic acid, fumaric acid, 3-methyl-2-pentendioic acid),
erythrose (a triol compound), 2,3-dihydroxypropanoic acid (a dihydroxy acid), hydroxy-diacids (2-
hydroglutaric acid, 2-hydroxy-4-isopropylhexandioic acid, 3-hydroxy-2-pentenedioic acid, 3-hydroxy-
3-methylglutaric acid, 3-hydroxyhexandioic acid, malic acid) and also 2-MGA, 3-MBTCA and DHOPA.
Higher concentrations for DHOPA, 2-MGA, MBTCA, and HGA are observed from 20 to 24 July (see
supplementary material 2). 2-MGA is formed, in presence of $NO_x$ (Ding et al., 2014, Fu et al., 2009;
Giorio et al., 2017), through the oxidation of methacrolein and methacrylic acid, both oxidation
products of isoprene. This period is characterized by the highest $NO_x$ concentrations of the campaign
(averaged concentrations of 1 ppbv against 0.6 ppbv for the rest of the campaign). Some dicarboxylic
acids (e.g. malonic acid, succinic acid and glutaric acid) also show a rise in their concentrations during
this period. This suggest a strong photochemical activity with an important aging of the air masses
collected and an advanced photochemical age for this period, also characterized by high OH missing
reactivity observed at the site (Zannoni et al., 2017). On the contrary, from the 27th of July to the end
of the campaign, levels of concentrations for these compounds decrease (see supplementary material
2) suggesting less aged air masses. This is also revealed by the higher (cis-pinonic acid + pinic
acid)/MBTCA ratio observed during this last period (see supplementary material 2). Indeed, this ratio
allows the evaluation of the oxidation state of air masses since cis-pinonic acid and pinic acid are first
generation oxidation products of monoterpenes while MBTCA is known to be a higher generation
oxidation product (Ding et al., 2014).
Observations of MSA (methanesulfonic acid, $CH_3SO_3H$) and water soluble HULIS are reported in
supplementary material S3. MSA is an oxidation product of dimethyl sulfide (DMS), a gaseous
compound emitted by marine phytoplankton activity, and is mostly present in particulate phase. MSA



can therefore be used to identify influence of marine chemistry on aerosol composition. Higher MSA
concentrations are observed on 23 to 28 July and on 4 August when air masses were coming from the
west sectors and spent days above sea (see Michoud et al., 2017) and on the first period of the
campaign (15-18 July) when air masses were stagnant with very low wind speed (see Michoud et al.,
2017). In summer, HULIS are mostly formed through secondary oligomerization processes in the
particulate phase (Baduel et al., 2010). Higher water soluble HULIS concentrations are observed on 20-
21 July when air masses are originating from north-east sector bringing continental aged air-masses
(Michoud et al., 2017) and on 27 July when air masses were coming from the southern sector with
large biogenic influence (Michoud et al. 2017). This is consistent with the formation of HULIS through
secondary oligomerization processes in summer from both anthropogenic and biogenic precursors
(Srivastava et al., 2018).
**3.2.4 Molecular characterization of particulate matter**
A time series of total mass quantified by TD-GC/MS in $PM_{2.5}$ is presented in Figure 9. This sum has been
calculated using the QL/2 (quantification limit/2) value when data were below the limit of
quantification. The sum of all the compounds measured by TD-GC/MS represents an average of
630 ng $m^{-3}$ for the whole campaign with a minimum of 54 ng $m^{-3}$ and a maximum of 2400 ng $m^{-3}$
measured on the 17[th] of July.
This sum is also compared to the organic matter mass concentration in $PM_{2.5}$ (see Figure 9). OM is
calculated using the organic carbon (OC) concentration measured by the SUNSET field instrument with
a ratio between OC and OM of 1.9 for Cape Corsica as proposed by Michoud et al. (2017). On average
18% of the total OM mass can be explained by the compounds measured by TD-GC/MS for the whole
campaign. From 12 to 29 July, oxygenated compounds measured by TD-GC/MS represent more than
20% on average of measured OM while they represented less than 10% between July 29 and August
4. If measured water soluble HULIS are added to these compounds, analysed compounds represent
36% of measured OM on averaged and up to 100% on 16 July.
Some of the compounds identified and quantified by TD-GC/MS, especially carboxylic acids, are soluble
in aqueous phase. To allow a comparison between TD-GC/MS measurement and WSOC (Water Soluble
Organic Carbon) measurements conducted by PILS-TOC, only soluble compounds measured by TD-
GC/MS have been selected (see Figure 10). Indeed, we considered only the compounds having a
Henry's law constant higher than $10^4$ M $atm^{-1}$. For every compounds measured by TD-GC/MS, the
Henry's law constants have been determined by the Structure Activity Relationship (SAR) developed
by Raventos-Duran et al. (2010) using the online platform of GECKO-A model (Aumont et al., 2005;
http://geckoa.lisa.u-pec.fr/generateur_form.php). At the end, 39 different compounds have been
selected for the calculation of this sum and no aldehyde or ketone were kept in this selection.



Comparing the sums of compounds measured by TD-GC/MS considering only soluble ones or
considering all of them reveals very similar behaviors and level of concentrations (see Figure 10). On
average, soluble compounds represent 72% of the total concentration of PM measured by TD-GC/MS
despite the important number of compounds not considered as soluble (26 compounds over 58 not
considered). Time series of soluble compounds measured by TD-GC/MS and of WSOM have similar
behaviors with higher concentrations during the period comprised between 17 and 23 July and smaller
concentrations at the end of the campaign. It is worth noting that WSOM corresponds to $PM_1$ while
TD-GC/MS measurements concern $PM_{2.5}$. On average, the sum of the soluble compounds measured by
TD-GC/MS represented 24% of the total WSOM measured by PILS-TOC. If measured water soluble
HULIS are added to these soluble compounds, analysed water soluble compounds represent 58% of
measured WSOM on averaged and up to 100% on 15 and 17 July.
Time series and average composition of the $PM_{2.5}$ measured by TD-GC/MS are presented respectively
in Figure 11 and Figure 12. Almost half of the $PM_{2.5}$ measured by TD-GC/MS are characterized by di-
carboxylic acid (49%) with oxalic acid being the most important by far. Other contributors to $PM_{2.5}$
composition measured by TD-GC/MS are tri-carboxylic acids (15%), alcohols (13%), aldehydes (10%),
di-hydroxy-carboxylic acids (5%), monocarboxylic acids and ketones (3% each) and hydroxyl-carboxylic
acids (2%). High concentrations of di-carboxylic acids are observed from 13 to 28 July (441 ng m$^{-3}$ on
average; 51% of the total OM measured by TD-GC/MS). After the 29[th] of July, the contribution of di-
carboxylic acids decreases significantly to reach 30%. The end of the campaign is characterized by
intense fresh local biogenic emissions leading to less processed air masses and OM composed mostly
by mono-functionalized compounds. On a general basis, organic acids constitute the principal
contributors to the fraction of organic aerosol measured by TD-GC/MS during this campaign while only
few chemical processes are known to lead to their formation (see section 3.2.3). The identification of
many di-carboxylic acids implies the existence of unknown chemical processes both in gaseous phase
and even more probably in particulate phase to explain their formation (Hammes et al., 2019). These
missing processes in chemical mechanism included in models might contribute to their inability to
reproduce correctly the formation and aging of SOA. If HULIS are considered in this analysis, they
represent 59% of the total identified OM mass on average, ranging from 21% of contribution at the
beginning of the campaign to more than 80% at the end of the campaign (from 31 July to 3 August).
**3.2.5 Partitioning of organic carbon between gaseous and particulate phases**
Many of the compounds identified during the campaign are present in both the gas and aerosol phases.
The partitioning coefficient is therefore key to understand processes governing the equilibrium
between both phases. For the compounds present in both phases, an experimental partitioning
coefficient can be determined following eq. 2 relying on the Pankow equilibrium.



$$K_{pe,i} = \frac{F_i/TSP}{A_i} \qquad (2)$$

$K_{pe,i}$ corresponds to the experimental partitioning coefficient for the compounds i, $F_i$ corresponds to
the concentration in the particulate phase, $A_i$ corresponds to the concentration in gaseous phase and
TSP (Total Suspended Particulate matter) corresponds to the total mass concentration of particles
measured by TEOM-FDMS ($\mu$g m$^{-3}$). Uncertainties for experimental partitioning coefficients take into
account uncertainties on the measurement of concentrations in both phases (see section 2.3.5) and
on the TEOM measurement (estimated to be 25%).
Further, another expression of the Pankow equilibrium allows for the determination of theoretical
partitioning coefficients using eq. 3.

$$K_{pt,i} = \frac{760RTf_{om}}{MW_{om}\zeta_i 10^6 p^0_{L,i}} \qquad (3)$$

$K_{pt,i}$ corresponds to the theoretical partitioning coefficient for the compounds I, R to the ideal gas
constant, T to the temperature in Kelvin, $f_{om}$ to the OM mass fraction, $MW_{om}$ to the averaged molar
mass of compounds constituting organic particulate matter(g mol$^{-1}$), $\zeta_i$ to the activity coefficient, $p^0_{L,i}$
to the saturation vapor pressure (Torr). Saturation vapor pressures have been determined at 295K
(averaged temperature of the campaign) using three different models (Moller et al., 2008; Myrdal and
Yalkowsky, 1997; Nannoolal et al., 2008). $f_{om}$ has been set to 0.8 using the averaged OC/TC ratio
measured by the SUNSET field instrument.
Experimental (averaged over the campaign) and theoretical partitioning coefficients obtained for
compounds identified in both phases are presented in Table 4 and Figure 13 and are compared to
experimental coefficient obtained in a previous field study in Corsica and a chamber study in the
EUPHORE simulation chamber (Rossignol et al., 2016). For most of the compounds, experimental
partitioning coefficients obtained for the three campaigns are relatively close to each other, with some
differences that can however reach up to an order of magnitude (e.g. dimethylglyoxal or acrolein, even
two orders of magnitude for glyoxal). These observed differences are small compared to the
differences recorded between experimental and theoretical coefficients, with an observed
underestimation of theoretical coefficients varying from 1 to 7 orders of magnitude. It is worth noting
that the three models used for theoretical coefficients determination are in good agreement. Higher
differences between experimental and theoretical coefficients are observed for hydroxyl compounds
and carboxylic acids with a shift of the equilibrium toward the particulate phase for experimental
partitioning coefficients. It is worth noting that a denuder is used upstream the filter collection to avoid
overestimation of particulate organic matter due to adsorption of semi-volatile compounds onto the



filter, therefore excluding potential positive artefact for concentrations of compounds in particulate
phase that could have led to overestimation of experimental partitioning coefficients. Furthermore,
underestimation of gaseous concentrations for these compounds in such high proportion is unlikely,
especially when we look at the comparisons performed for OVOCs with other measurement
techniques (see section 3.2.2.1).
The differences observed between experimental and theoretical partitioning coefficient may be
explained by the high humidity conditions encountered during the campaign (mean RH value of 70%,
see Table 3). Indeed, theoretical partitioning coefficient as described by the Pankow equilibrium does
not take into account the presence of an aqueous phase or a deliquescent aerosol, while, soluble
organic compounds can split between gaseous, aqueous and particulate phase. Concerning the
partitioning between the gaseous and aqueous phases, the Henry law's constant and the activity
coefficients are considered to calculate the thermodynamic equilibrium.
These differences could also be explained by the fact that the equilibrium between both phases is not
reached. This could be due to the viscosity of particles. Some studies showed that organic aerosol can
be found in various states, from liquid to semi-solid (viscous) (Bateman et al., 2016; Booth et al., 2014,
Shiraiwa et al., 2011; Virtanen et al., 2010). The viscosity of the particle can limit the diffusion inside
the particle, which can lead to an inhomogeneity in the composition with the formation of a gradient
of concentrations between the surface and the center of the particle (Chan et al., 2014; Davies and
Wilson, 2015; Zobrist et al., 2011). The equilibrium could therefore only concern an external layer of
the particle and the gaseous phase (Davies and Wilson, 2015); or on the contrary a semi-solid external
layer, caused by the aging of the particle, could prevent the equilibrium to settle between the
particulate bulk and the gaseous phase.
Furthermore, Soonsin et al. (2010) showed that the physical state of the particle can influence the
activity coefficient of some compounds and especially of dicarboxylic acids. Partitioning coefficients
are calculated considering a liquid phase for aerosols. Considering a solid or semi-solid phase for
aerosols would lead to a decrease in the vapor pressure estimation for such compounds and therefore
to higher theoretical partitioning coefficients.
In addition, polymerization and oligomerization processes in the particulate phase have been
highlighted in previous studies through the identification of compounds with high masses (Hallquist et
al., 2009; Kalberer et al., 2004; Lim et al., 2010; Tolocka et al., 2004). The formation of oligomers
increases the viscosity of the particle during its aging (Abramson et al., 2013). These reactions could
also explain the presence of semi-volatile compounds in the particulate phase in such high proportion,
especially for carbonyls that have high vapor pressure and which should not be detected in the aerosol





phase based on the theory. Indeed, numerous studies reveal the possibility of formation of oligomers,
inside the particle, from carbonyls such as α-di-carbonyls, for example glyoxal or methylglyoxal (Gao
et al., 2004a, 2004b; Hastings et al., 2005; Iinuma et al., 2004; Jang et al., 2002, 2003; Jang and Kamens,
2001; Liggio et al., 2005a, 2005b; Lim et al., 2010; Tolocka et al., 2004). These reactions are favored
under low water content in the particles. On the contrary, under higher humidity conditions, oligomers
can form back monomer compounds which in case of viscous particle can be trapped into the
particulate phase. It is worth noting that higher experimental partitioning coefficients are found for
most compounds on 20 July and 26-27 July while water soluble HULIS concentrations are at their
maximum. HULIS are known to be formed through secondary oligomerization processes in summer
(Baduel et al., 2010), supporting the hypothesis that these kind of processes might be partly
responsible for the disagreement between experimental and theoretical partitioning coefficient.
Even if an analytical artifact cannot be ruled out, for example a fragmentation of oligomers to form
back the monomer compounds during the analysis, numerous evidences support the experimental
results presented here and suggest that the instantaneous equilibrium being established between
gaseous and particulate phases assuming a homogeneous non-viscous particle phase is not fully
representative of the real atmosphere.
**Conclusion**
A multiphasic molecular characterization of oxygenated compounds has been carried out during the
ChArMEx SOP 1b field campaign held in Ersa Corsica during July 2013 using an analytical technique
based on multi-support sampling (filters and adsorbent containing cartridges), derivatization
procedure and TD-GC/MS analysis. The deployment of this analytical technique in the field allows the
identification of 97 different compounds in the gas (24 different compounds) and aerosol (50 different
compounds) phases, some of them being present in both phases (23 different compounds). These
compounds include simple carbonyls, alcohols or carboxylic acids as well as multi-functional
compounds up to four functional groups. Among all the quantified compounds, the important
contribution of organic acids (67% of the organic aerosol concentration measured by TD-GC/MS)
emphasis the existence of unknown chemical processes both in the gaseous phase and even more
probably in the particulate phase to explain their formation. The absence of such processes in chemical
mechanisms may contribute to the inability of models to correctly reproduce the formation and aging
of SOA.
Comparisons of these measurements with other measurements performed at the site when available
reveal fair agreement on the whole for almost all compounds experiencing redundant measurement
in both phase with concentrations at least in the same order of magnitude. Noticeable disagreements



(larger than a factor of 8 and up to a factor of 15) have however been found for glyoxal in the gas phase
between TD-GC/MS measurements and DNPH cartridges analysis and for malic and succinic acid in the
particulate phase between TD-GC/MS measurements and HPLC analysis. Nevertheless, comparisons
of TD-GC/MS with other techniques deployed during the campaign are in general agreement,
validating their use to conduct further analysis.
While the data obtained are very valuable to provide additional insight into the composition of organic
matter for air masses encountered during the campaign, it is worth noting that it represents only a
fraction of the total mass of organic matter. Indeed, an attempt to close the mass budget of organic
aerosol using the TD-GC/MS measurements reveal that the sum of all particulate oxygenated organic
compounds measured by this technique account for 18% of the total OM mass on average for the
whole campaign. This portion of OM identified at the molecular scale is not constant and mostly
depends on the oxidation state of the sampled air masses. If we only consider the soluble compounds
measured by TD-GC/MS, they represent 24% of the total WSOM on average. Therefore, a sizeable
fraction of the OM mass was identified by TD-GC/MS analysis, but a very large fraction of OM mass
remained unidentified during the campaign, highlighting the complexity of an exhaustive
characterization of the OA chemical composition at the molecular scale. An important fraction of this
unidentified OM mass is due to HULIS.
Finally, for the compounds quantified in both the gas and the aerosol phases, a comparison between
experimental and theoretical partitioning coefficients has been performed revealing in most cases a
large underestimation by the theory reaching 1 to 7 orders of magnitude. It indicates that the
partitioning theory is most often inappropriate, since it is based on the instantaneous equilibrium
being established between gaseous and particulate phases, assuming a homogeneous non-viscous
particle phase, which is the base for aerosol modeling. Furthermore, the partitioning of semi-volatile
compounds is influenced by meteorological conditions (humidity, temperature) and inherent
properties of particles (viscosity, water content, organic fraction concentrations, acidity, etc.). In
addition, the way these conditions impact the partitioning of semi-volatile compounds strongly
depends on the physico-chemical properties of the considered compounds (solubility, saturation vapor
pressure, reactivity, etc.).
**Data availability.**
Access to the data used for this publication is restricted to registered users following the data and
publication policy of the ChArMEx program (http://mistrals.sedoo.fr/ChArMEx/ Data-
Policy/ChArMEx_DataPolicy.pdf).



**Author contributions.**

VM and EH participated in the field campaign and prepared the paper with inputs from all co-authors.
LC, ELG and JFD were involved in TD-GC/MS measurements and supervised this work. SD, IF, TL, NL
and SS participated in the field campaign and were in charge of VOC measurements (GC-FID/MS, PTR-
MS, Active sampling on DNPH cartridges). AC and FG were in charge of inorganic trace gases
measurements (NOx and $O_3$). JS participated in the field campaign and was in charge of aerosol
measurements by ACSM, OCEC instrument, PILS-TOC and IC. JLJ and NM were in charge of aerosol
speciation measurements during the campaign through filter analysis (IC, GC/MS, HPLC, HULIS
measurements).

**Competing interests.**

The authors declare that they have no conflict of interest.

**Special issue statement.**

This article is part of the special issue "CHemistry and AeRosols Mediterranean EXperiments (ChArMEx;
ACP/AMT inter-journal SI)". It does not belong to a conference.

**Acknowledgements.**

This study received financial support from the MISTRALS and ChArMEx programs, ADEME, the French
Environmental Ministry, and the Communauté Territoriale de Corse (CORSiCA project). This project
was also supported by the CaPPA project (Chemical and Physical Properties of the Atmosphere),
funded by the French National Research Agency (ANR) through the PIA (Programme d'Investissement
d'Avenir) under contract ANR-11-LABX-0005- 01 and by the Regional Council Nord-Pas de Calais and
the European Funds for Regional Economic Development (FEDER).

The authors also want to thank Eric Hamonou and François Dulac for logistical help during the
campaign and all the participants of the ChArMEx SOP1b field campaign. This paper is dedicated to the
memory of our friend and colleague Laura Chiappini, who passed away shortly after the campaign.
Laura conceived the original idea for this work and created the conditions to have this experimental
worked done. Analyses at IGE were performed on the Air O Sol plateform partly funded with the Labex
OSUG@2020 (ANR10 LABX56)





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


1    Table 1: Thermal desorption method and GC/MS parameters

| | | |
|---|---|---|
| **Thermal desorption parameters for samples** | temperature | 300°C |
| | time | 15 minutes |
| | flow | 50 mL min$^{-1}$ |
| | split flow | No split flow |
| **Thermal desorption parameters for the trap** | Temperature | From -10°C to 300°C |
| | Time | 12 minutes |
| | flow | 10 mL min$^{-1}$ |
| Temperature of transfer lines | | 200°C |
| **GC Parameters** | Carrier gas | He |
| | Carrier gas flow | 1 mL min$^{-1}$ |
| | Temperature gradient | 40°C / 10°C min$^{-1}$ / 305°C (10 min) |
| | Split flow | 0.2 mL min$^{-1}$ |
| | Transfer line temperature to MS | 305 °C |
| **MS parameters** | Scan m/z | 40 to 800 |
| | Solvent delay | 5 min |
| | Quadrupole temperature | 150°C |
| | EI -Source temperature -Ionization Energy | 230°C 70 eV |
| | CI -Source temperature -Reagent gas -Ionization Energy | 250°C CH$_4$ 50 eV |





1     Table 2 : List of substitutes used for internal calibration

| Substitutes used for carbonyl compounds | Substitutes used for hydroxyl compounds |
|---|---|
| 3-methylbutanal-d2 | Pentanoic acid-d9 |
| Butanal-d8 | Heptanoic acid-d13 |
| 4-methyl-2-pentanone-d5 | Succinic acid-d4 |
| Benzaldehyde-d6 | 2-methyl-d3-2-propyl-1,3-propanediol |
| Acetophenone-d8 | Glycerol-d8 |
| 2-hexanone-d5 | Tartaric acid-2,3-d2 |
| 2,3-butanedione-d5 | |
| 2,5-hexanedione-d10 | |





1    Table 3: meteorological conditions, environmental parameters and mass concentrations of $PM_{10}$, $PM_1$

2    and organic fraction in $PM_1$ during the ChArMEx campaign at ERSA

| Meteorological and Environmental Parameters | Mean | Median | Max | Min |
|---|---|---|---|---|
| Temperature (°C) | 23 | 23 | 32 | 19 |
| Relative Humidity (%) | 70 | 73 | 100 | 27 |
| Wind Speed (m s$^{-1}$) | 3.6 | 3.1 | 13.2 | - |
| $O_3$ (ppbv) | 65 | 65 | 111 | 42 |
| $NO_x$ (ppbv) | 0.57 | 0.45 | 4.93 | 0.06 |
| Mass concentrations (µg m$^{-3}$) | Mean (±1σ) | Median | Max | Min |
| $PM_{10}$ | 12 (±4.8) | 12 | 31 | 2 |
| $PM_1$ | 8.4 (±4.4) | 8.4 | 22 | 0.2 |
| Organic fraction ($PM_1$) | 3.7 (±1.7) | 3.5 | 8.1 | 0.2 |





Table 4: Experimental (averaged over the campaign with ±XX% representing 1σ standard deviation
over the campaign) and theoretical partitioning coefficients determined for this study and compared
to previous field and chamber campaigns.

| | This study | Corsica [a] | EuPhoRe [a] | Kpt,i MOL [b] | Kpt,i NAN [c] | Kpt,i MYR [d] |
|---|---|---|---|---|---|---|
| Propanal | $6.1\times10^{-3} \pm 75\%$ | $2.2\times10^{-3} \pm 50\%$ | | $2.6\times10^{-10}$ | $2.6\times10^{-10}$ | $4.7\times10^{-10}$ |
| Pentanal | $6.5\times10^{-4} \pm 106\%**$ | $1.8\times10^{-4} \pm 51\%$ | | $3.2\times10^{-9}$ | $3.2\times10^{-9}$ | $3.8\times10^{-9}$ |
| Hexanal | $1.3\times10^{-3} \pm 61\%$ | | | $1.0\times10^{-8}$ | $1.0\times10^{-8}$ | $1.1\times10^{-8}$ |
| Heptanal | $5.1\times10^{-4} \pm 91\%$ | | | $3.3\times10^{-8}$ | $3.2\times10^{-8}$ | $3.4\times10^{-8}$ |
| Acrolein | $7.3\times10^{-4} \pm 74\%$ | $6.1\times10^{-3} \pm 50\%$ | | $3.6\times10^{-10}$ | $3.6\times10^{-10}$ | $3.7\times10^{-7}$ |
| Methacrolein | $7.3\times10^{-4} \pm 69\%$ | | | $7.2\times10^{-10}$ | $7.2\times10^{-10}$ | $9.0\times10^{-10}$ |
| Methyl Vinyl ketone | $5.8\times10^{-4} \pm 57\%$ | | | $1.3\times10^{-9}$ | $1.3\times10^{-9}$ | $5.6\times10^{-10}$ |
| Nopinone | $5.5\times10^{-4} \pm 53\%$ | | | $1.7\times10^{-7}$ | $1.7\times10^{-7}$ | $1.9\times10^{-7}$ |
| Dimethylglyoxal | $5.0\times10^{-3} \pm 65\%$ | $5.6\times10^{-4} \pm 70\%$ | $6.2\times10^{-4} \pm 47\%$ | | $3.4\times10^{-9}*$ | $7.0\times10^{-9}*$ |
| Methylglyoxal | $3.6\times10^{-3} \pm 60\%$ | $2.2\times10^{-2} \pm 132\%**$ | $1.3\times10^{-3} \pm 84\%$ | | $8.6\times10^{-10}*$ | $2.1\times10^{-9}*$ |
| Levulinic acid | $5.1\times10^{-3} \pm 77\%$ | | | $1.7\times10^{-5}$ | $4.4\times10^{-6}$ | $2.9\times10^{-6}$ |
| Methacrylic acid | $1.5\times10^{-4} \pm 198\%**$ | | | $8.4\times10^{-8}$ | $7.6\times10^{-8}$ | $8.9\times10^{-8}$ |
| Glycolic acid | $3.1\times10^{-2} \pm 268\%**$ | | | $8.5\times10^{-5}$ | $1.3\times10^{-5}$ | $2.0\times10^{-6}$ |
| Glycerol | $1.1\times10^{-2} \pm 62\%$ | | | $7.1\times10^{-4}$ | $8.4\times10^{-4}$ | $1.3\times10^{-5}$ |

[a] Rossignol et al., 2016; [b] Moller et al., 2008 (coupled with Nannoolal et al. (2004) method for boiling point determination) ; [c] Nannoolal et
al., 2008 (coupled with Nannoolal et al. (2004) method for boiling point determination) ; [d] Myrdal and Yalkowsky, 1997 (coupled with
Nannoolal et al. (2004) method for boiling point determination)
* Coefficients extracted from Rossignol, 2012 at temperature of 300 K other parameter (MW$_{om}$ et ζ$_i$) kept similar.
** Partitioning coefficients are comprised between 0 and 1. Experimental uncertainties greater than 100% mean that the experimental
value is comprised between 0 and more than twice its values.

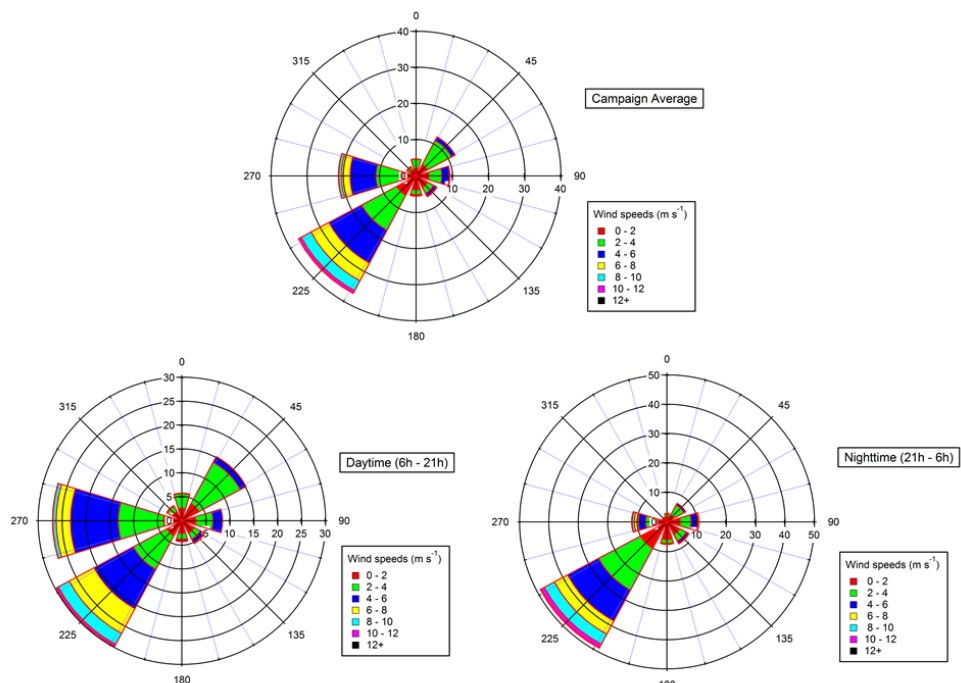

2    Figure 1 : Wind roses from July 15th to August 5th 2013 (top panel), during daytime only (bottom left

3    panel) and during nighttime only (bottom right panel). Wind direction is expressed in ° and radial axe

4    express the relative occurrence of wind in each 30° sector (%).



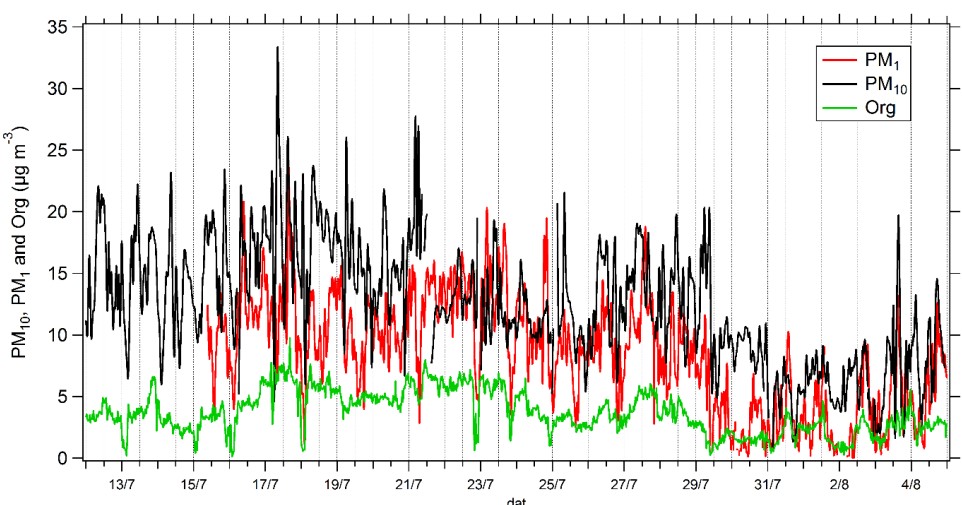

2      Figure 2 : Time series of mass concentrations of PM$_{10}$ (black line), PM$_1$ (red line) and organic fraction

3      in NR-PM$_1$ (green line).

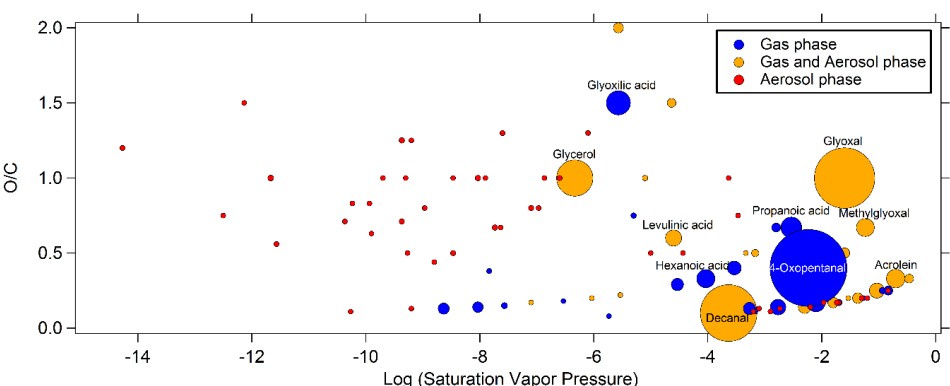

Figure 3: Distribution of compounds identified by TD-GC/MS during the ChArMEx campaign according to the logarithm of their saturation vapor pressure (horizontal axis) and of their O/C ratio (vertical axis). The phase in which they are detected is color-coded: blue for compounds only detected in the gas phase, red for aerosol phase only and orange for compounds detected in both phases. Each dot represents a single compound and the dot area is proportional to the sum of concentrations if detected in both phases from 0.3 ng m$^{-3}$ for the smallest dot to 3.9 μg m$^{-3}$ for the biggest one. Name of some noticeable compounds are also given.





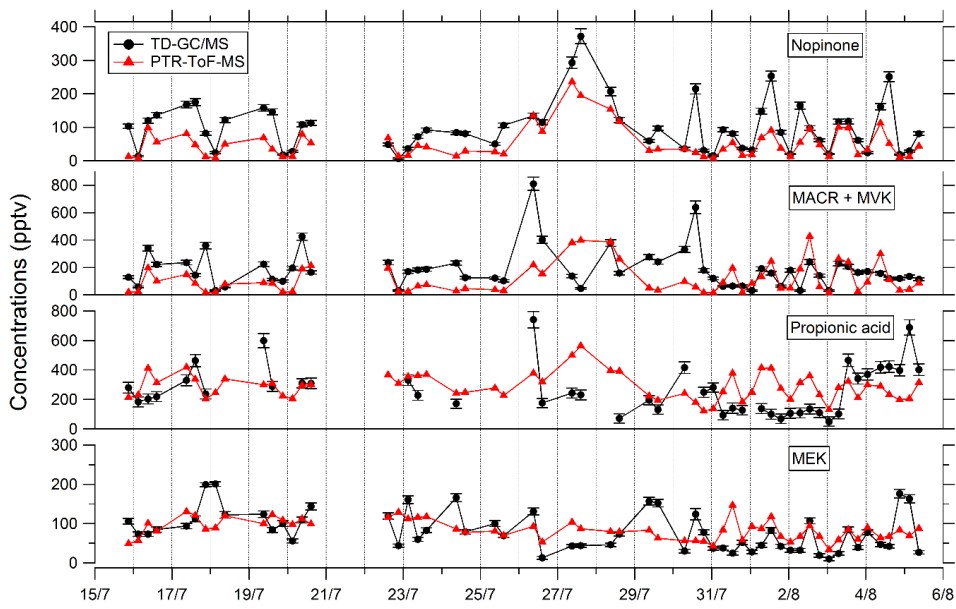

Figure 4 : Comparison of ATD-GC-MS data with PTR-ToF-MS data averaged over the same time step for
nopinone, the sum of methacrolein and methyl vinyl ketone, propionic acid and methyl ethyl ketone.
Error bars correspond to the 1σ uncertainties of TD-GC/MS measurements. Error bars correspond to
the 1σ uncertainties of TD-GC/MS measurements.




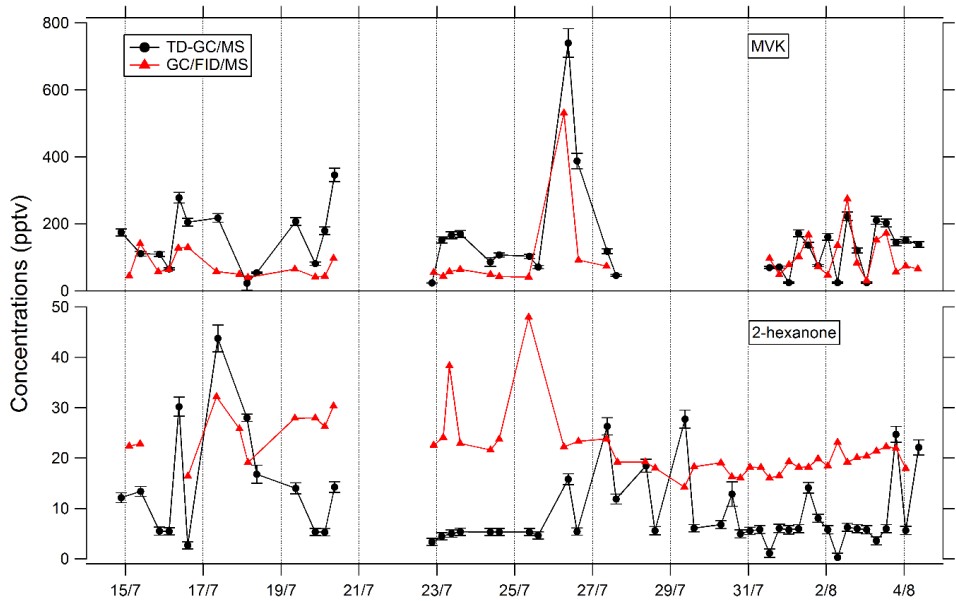

2   Figure 5 : Comparison of ATD-GC-MS data with GC/FID/MS data averaged over the same time step for

3   methyl vinyl ketone and 2-hexanone. Error bars correspond to the 1σ uncertainties of TD-GC/MS

4   measurements. Error bars correspond to the 1σ uncertainties of TD-GC/MS measurements.

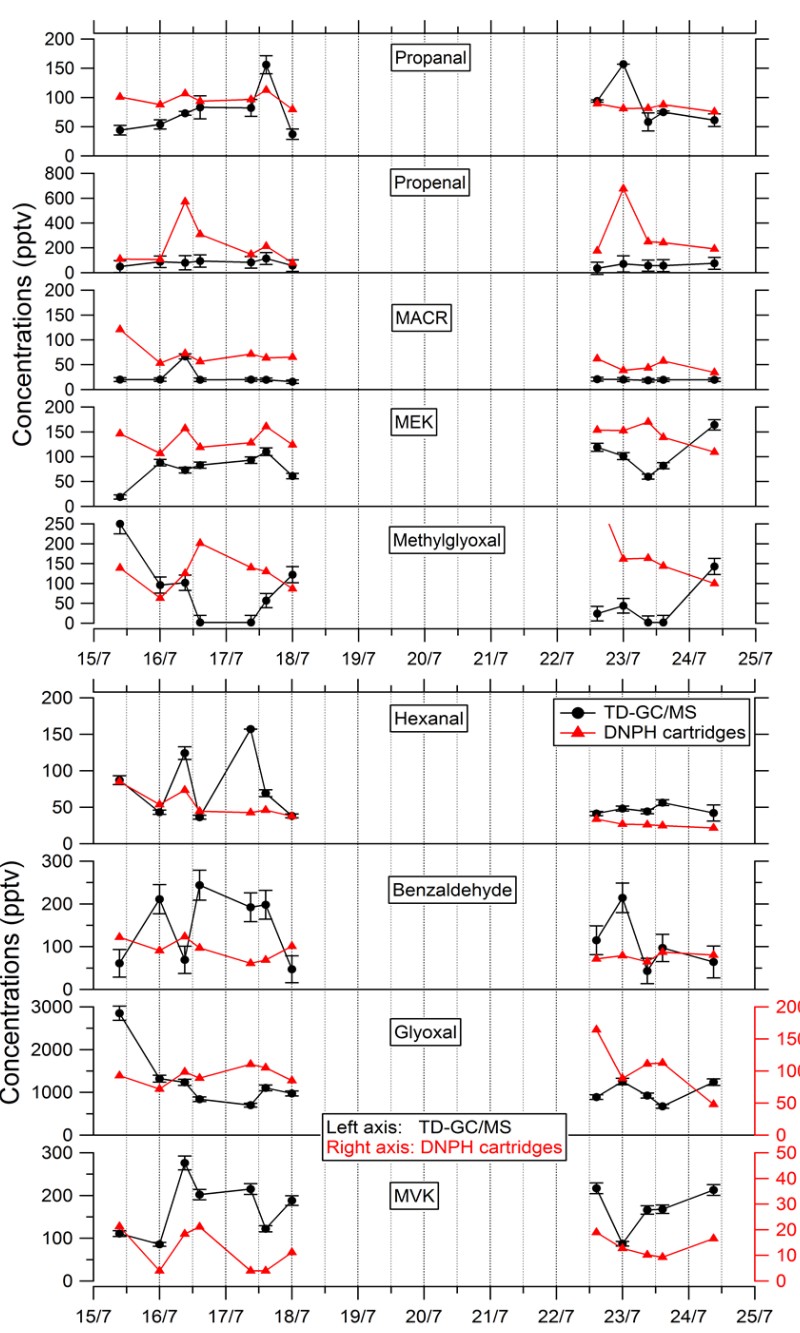

2 Figure 6 : Comparison of ATD-GC-MS data with DNPH cartridges analysis for 9 OVOCs. Error bars

3 correspond to the 1σ uncertainties of TD-GC/MS measurements. Error bars correspond to the 1σ

4 uncertainties of TD-GC/MS measurements.

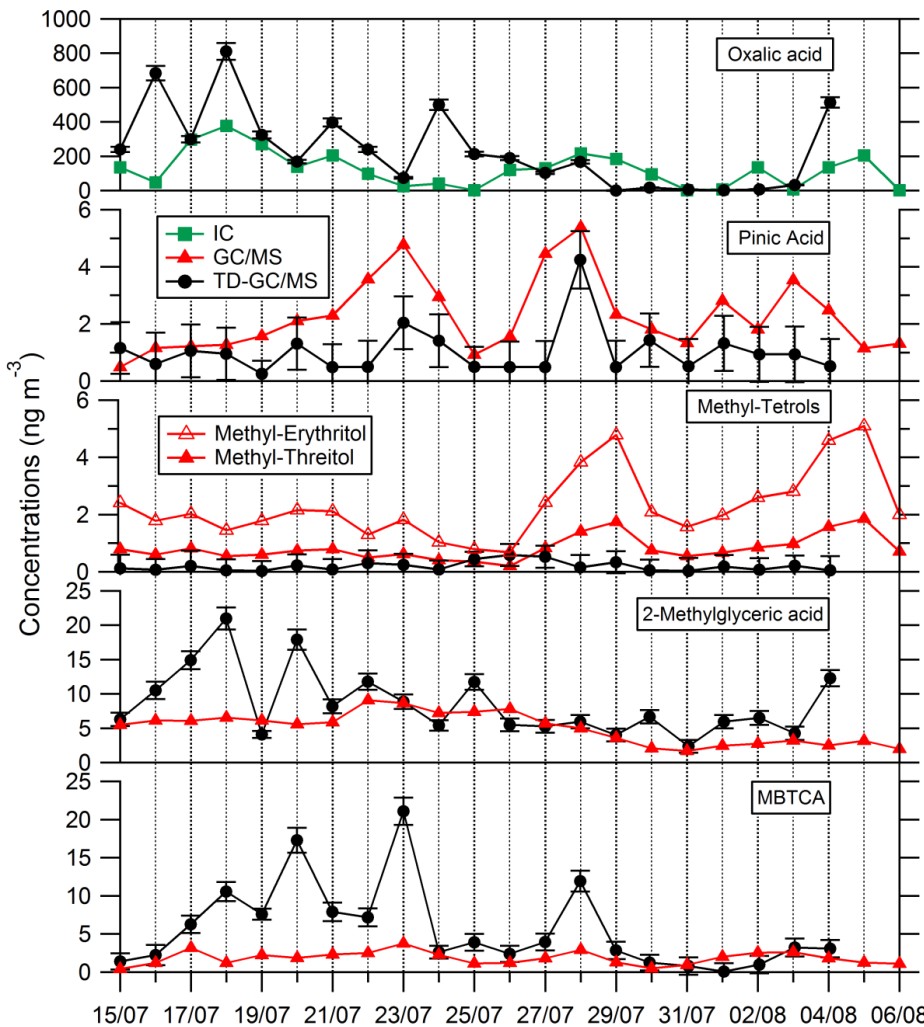

Figure 7 : Comparison of ATD-GC-MS data with ion chromatography and GC/MS analysis for particulate
oxalic acid, pinic acid, methyl tetrols, 2-methylglyceric acid and MBTCA (3-Methyl-1,2,3-tricarboxylic
acid). Error bars correspond to the 1σ uncertainties of TD-GC/MS measurements. Error bars
correspond to the 1σ uncertainties of TD-GC/MS measurements.

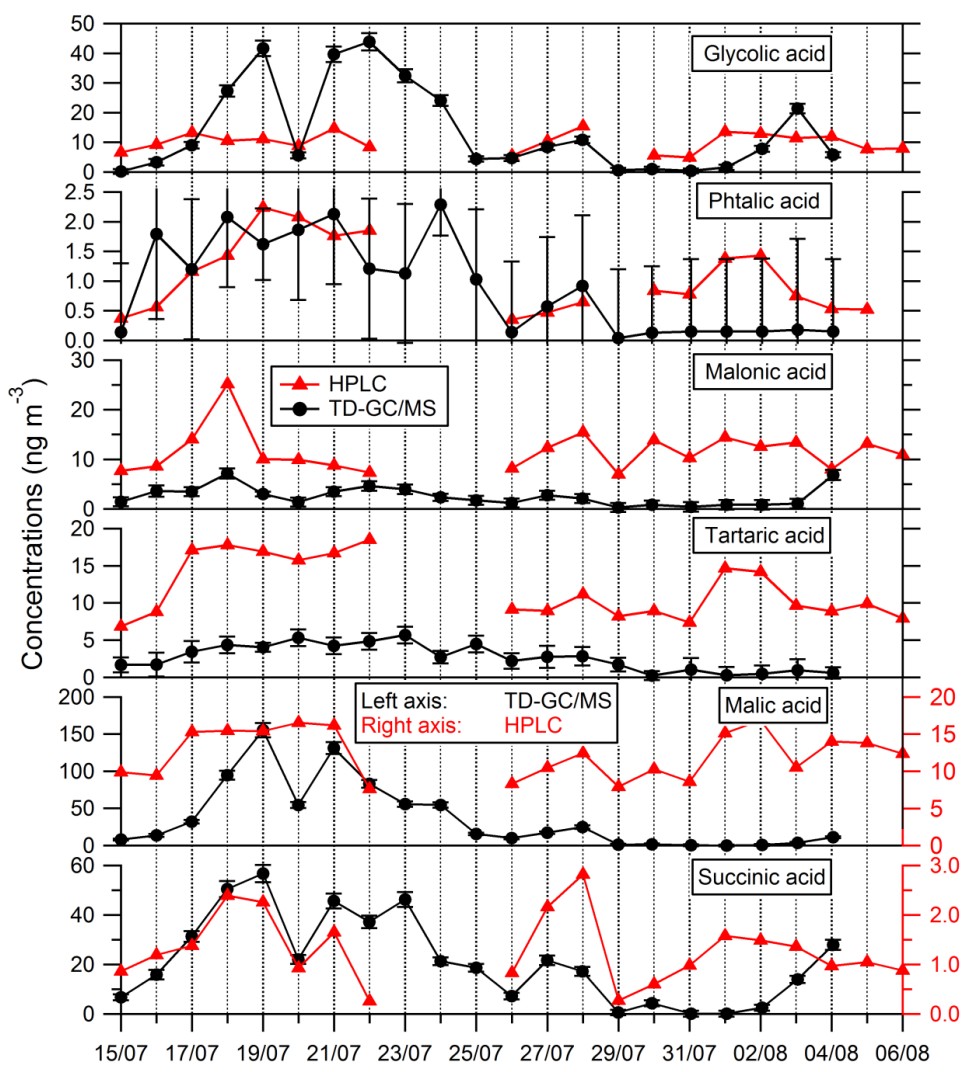

Figure 8: Comparison of ATD-GC-MS data with HPLC analysis for particulate glycolic acid, phtalic acid,
malonic acid, tartaric acid, malic acid and succinic acid. Error bars correspond to the 1σ uncertainties
of TD-GC/MS measurements. Error bars correspond to the 1σ uncertainties of TD-GC/MS
measurements.



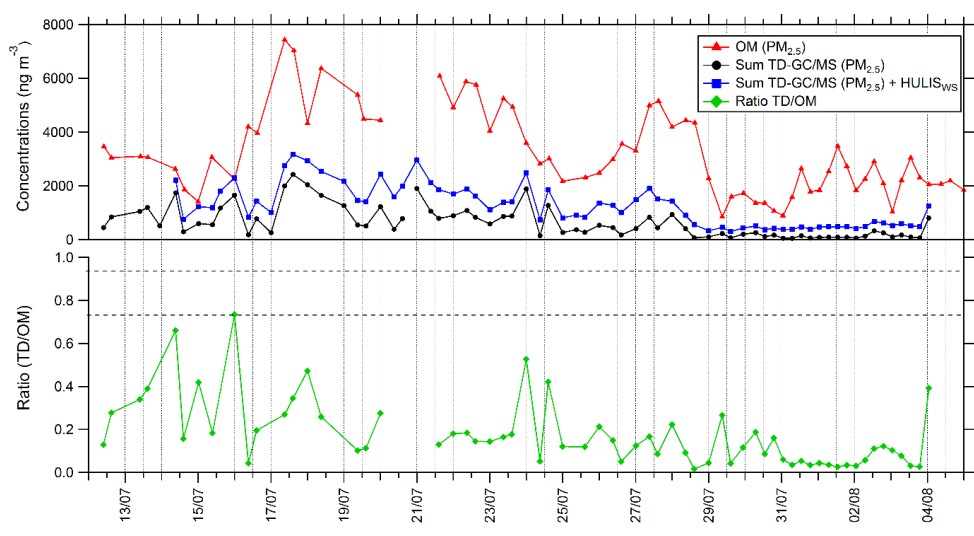

Figure 9: Time series of organic matter in PM$_{2.5}$ (red line), total sum of PM$_{2.5}$ from TD-GC/MS analysis (black line), total sum of PM$_{2.5}$ from TD-GC/MS analysis and water soluble HULIS analysis (blue line), and ratio of these two measurements (green line).

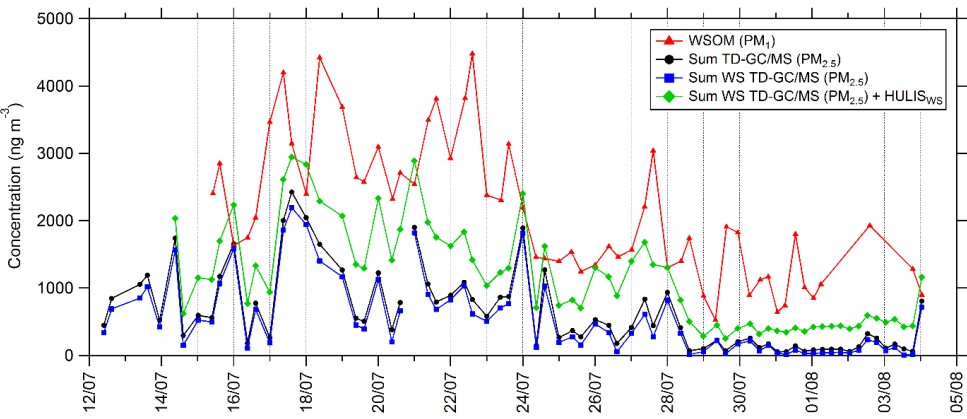

Figure 10: Time series of $PM_1$ water soluble organic matter (WSOM; red line), total sum of $PM_{2.5}$

measured by TD-GC/MS (black line), total sum of compounds measured by TD-GC/MS and having

henry's law constant higher than $10^4$ M atm$^{-1}$ measured by TD-GC/MS (WS TD-GC/MS, blue line), and

total sum of water soluble compounds measured by TD-GC/MS and water soluble HULIS (green line).





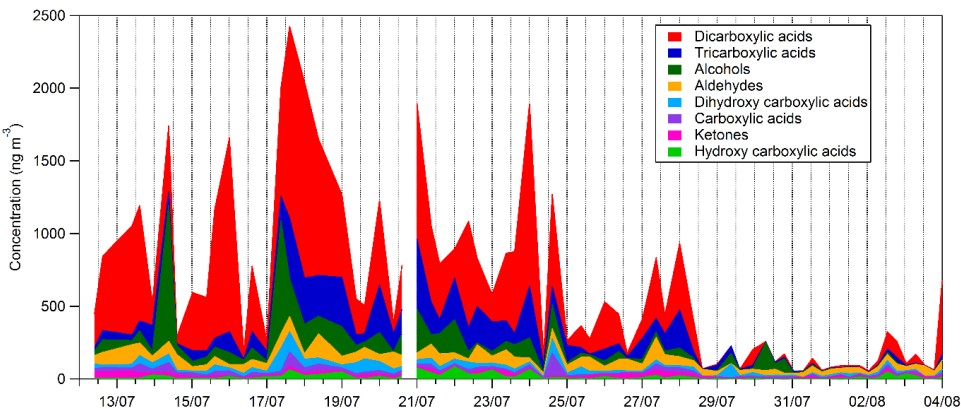

2    Figure 11: Time series of the composition of the sum of all compounds concentrations measured by

3    TD-GC/MS.





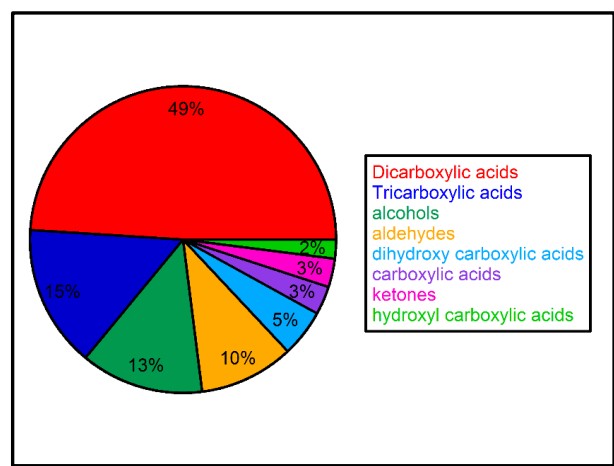

Figure 12: Campaign averaged relative composition of the sum of all compounds measured by TD-
GC/MS in the organic aerosol phase (hydroxyl-carboxylic acid-light green area, ketone-pink area,
carboxylic acid-purple area, dihydroxy carboxylic acid-light blue area, aldehyde-orange area, alcohol-
dark green area, tricarboxylic acid-dark blue area, dicarboxylic acid-red area).



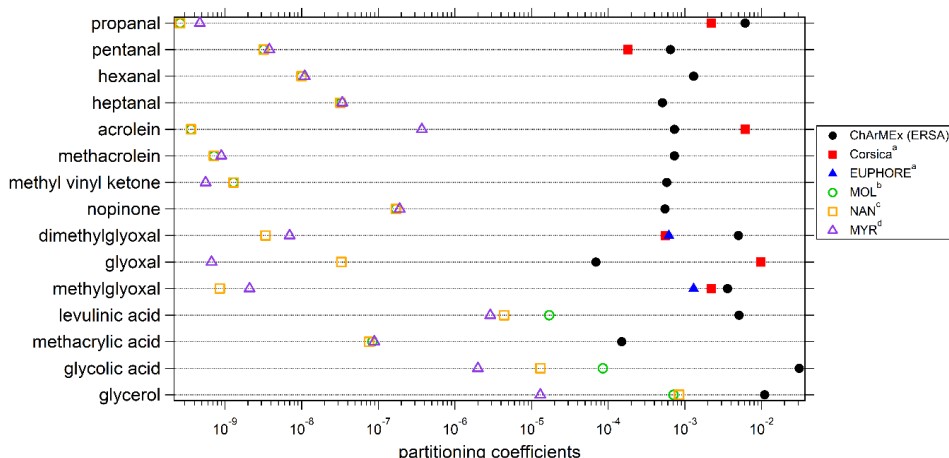

   [a] Rossignol et al., 2016; [b] Moller et al., 2008 (coupled with Nannoolal et al. (2004) method for boiling point determination) ; [c] Nannoolal et
   al., 2008 (coupled with Nannoolal et al. (2004) method for boiling point determination) ; [d] Myrdal and Yalkowsky, 1997 (coupled with
   Nannoolal et al. (2004) method for boiling point determination)
Figure 13: Experimental and theoretical partitioning coefficients determined for this study and
compared to previous field and chamber campaigns.