# Peer review of "Molecular characterization of gaseous and particulate"

_Atmospheric Chemistry and Physics, 2020_

## Referee Comment (RC1) · Anonymous Referee #1 · 26 Nov 2020

This study used online TD-GCMS, offline filter+GCMS to measure the polar organic compounds at both gas-phase and particle phase, and calculate their partitioning coefficients. The data is very valuable. However, I think the current manuscript is more like a data report, not a research article. I suggest changing the article type to Measurement Report. Nevertheless, there are still lots of deficiencies for this manuscript. The analysis is insufficient, and the manuscript is far from acceptance by ACP. A major revision is required for this manuscript. My suggestions are as following: In general, more analysis and discussion should be done in this work. e.g.

[Figure]

1) the partitioning behaviors of the compounds: I suggest comparing the theoretical calculation with the measurement, and to investigate whether T, RH, air mass can impact the partitioning coefficent.2) the sources of the quantified compounds, not only simply say the compounds are possibly from primary source or secondary formation. Although, as the authors claimed, some compounds' source are unclear, cluster analysis, PCA or PMF are still useful to analyze the source, even the source can be named as unknown source. At least we can understand what compounds are from the same or similar source. Detailed comments: Page 3, Line 19-20: TD-GCxGC-ToF/MS is the abbreviation of "Thermal desorption comprehensive two-dimensional gas chromatography–time of flight mass spectrometer". Notice that the expression of Gas ChromatographyxGC in line 20 is invalid. Page 4, Line 24: The expression of double chromatographic systems might be inappropriate and might be replaced by two-dimensional chromatography. Page 6. Line 7- 9: All of these measurements were used to assess the composition of organic carbon and to estimate the experimental partition coefficient of compounds measured in both phases to be compared with theoretical values. The expression of this sentence seemed to suggest that the present manuscript displayed a large amount of data while the aim of which was just the partitioning coefficient estimation. As the last sentence of the introduction part, it is confusing why this study just focused on the partitioning of these compounds while their sources and mechanisms remained unknown. Page 9, Line 17: the diameter of the column (250 $\mu$m in Line 17) might be 0.25 mm as this unit was widely used. Page 9, Line 29: why pentadecane was utilized as the internal standard? Page 19 Line 4: "than" to "as" Page 19 Line 9: It seems not reasonable that compounds in both gas phase and particle are high during day time. If it's controlled by the thermodynamic equilibrium, the compounds in the particle phase should be low during daytime due to evaporation. Page 19 Line 16-18, "For example…"unclear sentence Page 20 Line 1: "observations are different than for that" There are plenty of wrong expression and gramma mistake. I list only a small part of them. I strongly suggest the authors to carefully go through the text.

Please also note the supplement to this comment:
https://acp.copernicus.org/preprints/acp-2020-1051/acp-2020-1051-RC1-supplement.pdf

———————————————————

---

## Referee Comment (RC2) · Anonymous Referee #2 · 27 Nov 2020

The manuscript by Michoud et al. presents a method of analysis with TD-GC-MS of gas and particulate samples for the determination of oxygenated compounds. It compares the new method with alternative measurements in Cape Corsica and then uses data on the concentration of oxygenated compounds in the two phases to calculate partitioning coefficients. It finally compares the experimental values of partitioning coefficients with theoretical ones. I think that the manuscript could be published but after major revisions. I was also wondering whether an analytical chemistry journal might be more appropriate given that the methodological part is extensive. However, the study on

partitioning coefficients makes it absolutely interesting and suitable for an atmospheric chemistry journal.

Major comments:

Full validation of the new method is not reported. Please add LOD, LOQ, precision and recovery for each compound in both matrices. Please add all data pertaining to line 1-6 on page 10 to the supplementary materials. Please add an example chromatogram showing the separation of the different compounds and please report all retention times.

Concerning the method used for the calculation of VOC concentrations, not all VOCs react with water cluster ion and those who react may react through a ligand-switching mechanism rather than proton transfer. The formula used for the calculations is not generally applicable to all VOCs and each VOC should be treated correctly (reaction with H3O+ only, reaction with both H3O+ and H5O2+, proton transfer with H3O+ and ligand-switching with H5O2+). At line 30 page 10, the authors refer to the signal of the H3O+ ion. Is this measured on the isotope 18O and then corrected for the isotopic abundance? Was the 18O used also for measuring the signal of H5O2+?

What was the extraction protocol for DNPH cartridges?

Please add LOD, LOQ, precisions and recovery for the GC-MS method described at page 12-13.

IC and HPLC-PAD analysis are mentioned but methods are not reported.

Line 13 page 17 and elsewhere: may you be quantitative in discussing the agreement between methods rather than reporting qualitative adjectives such as "fair agreement"?

Line 31 page 17, was the thermodegradation observed with the standards?

Line 9-16 page 18, did you check recovery and matrix effects to investigate the causes of this large disagreement between the results from the different methods?

Concerning the section 3.2.5 on the partitioning of oxygenated compounds between the two phases, I have a general concern about the input data from the TD-GC-MS measurements. For some compounds, there was a clear disagreement between TD-GC-MS results and alternative methods (e.g. glyoxal), so I am not confident that the data quality was always good enough to obtain reliable results on partitioning constants. Concerning more specifically the calculations, it is mentioned that TSP concentrations are used but there was no mention on the paper about TSP measurements and all TD-GC-MS analysis were done on PM2.5. In equation 3, activity coefficients are used but there is no reference or table reporting which ones were used.

Minor comments and typos: The introduction is very long and could be more focused. For example, I would remove the paragraph from line 26 at page 3 to line 8 at page 4. It does not add much to the discussion. Line 6 page 7: did you check that the filter does not cause absorption of SVOCs? Line 21 page 10: how humid was the humid zero air? Line 18-20 page 12: why the calibration was performed only 2 months before and not on-site? Did you do any quality check? Why a collection efficiency of 0.5 was used rather than a composition-dependent one? Line 18-20 page 13, the elution gradient is missing. Line 23-25 page 15, how high the ratios? Was there any long-range transport of dusts that could have influenced these ratios? Line 14 page 21, are you referring to compounds that were in-between the LOD and LOQ? Line 29-30 page 21, I am not sure if the discussion around Henry's law constants is relevant here. The absolute solubility of a compound in water may be more relevant in this case if we are discussing a transfer from particulate into water rather than a gas-water equilibrium. Line 4 page 25, oligomer production can happen also at high relative humidity and in the aqueous phase. Line 27 page 25, I believe "emphasis" should be "emphasizes" in this sentence. English is generally okay although it could be improved in some places. Please translate from French into English all labels in the plots in supplementary 2. In the table in supplementary 1 please change "substitut" with "substitute".

---

## Author Comment (AC1) · 3 Feb 2021

First of all, we would like to thank warmly the reviewers for their comments on the manuscript. We did our best to address all the comments and modified the manuscript accordingly. We believe it has significantly improved the paper. The changes made to the revised manuscript are summarized below.

**Response to referee #1:**

*I think the current manuscript is more like a data report, not a research article. I suggest changing the article type to Measurement Report. Nevertheless, there are still lots of deficiencies for this manuscript. The analysis is insufficient, and the manuscript is far from acceptance by ACP.*

We certainly disagree with this comment as we believe that our article is much more than a measurement report. The paper includes a discussion and scientific exploitation of about 6 pages. It provides the identification of compounds at a molecular scale and description of their behavior during the campaign which are valuable and rare information on their own to characterize the chemical composition of organic matter. This kind of description at a molecular scale is unfortunately too rare to provide the needed constraints on explicit modeling for example. In addition, the article also comprises a study on the partitioning coefficient of organic carbon between gaseous and particulate phases which is to our point of view very valuable and allows going further in the analysis of the data. In particular, it brings into light a strong underestimation of some partitioning coefficients questioning strongly the relevance of a representation of semi-volatile SOA based on volatility properties only.

We admit that the description of the instruments and analytical method is long (about 8 pages) but we believe that this detailed description as well as the intercomparison sections (2 pages) are needed to ensure the good quality of the data we use for the latter analysis. For all these reasons, the manuscript is more than a measurement report but because its organization seems to be confusing and in order to take into account the reviewer's feeling we have re-organized the various sections to highlights the scientific discussion.

*1) the partitioning behaviors of the compounds: I suggest comparing the theoretical calculation with the measurement, and to investigate whether T, RH, air mass can impact the partitioning coefficent.*
We do not understand the comment of the reviewer. The requested comparisons and investigation have been made. Is this a consequence of the initial lack of visibility of our discussion section? Anyhow, theoretical calculation are compared with measurements in the manuscript and discussed in detail section 3.2.5 (now section 4.3).
Concerning the impact of environmental parameters on partitioning, this is also discussed: We have chosen to only present the averaged results in table 4 and figure 13 for clarity and to avoid reaching a too important size for the article already quite long but the effects of air mass origin and composition (see for example lines 7-11 of page 25), and of humidity (see lines 6-12 of page 24) on the comparison are indeed discussed. Temperature dependence of partitioning coefficient is already included in the calculations and we therefore do not discuss further this aspect in the manuscript.

*2) the sources of the quantified compounds, not only simply say the compounds are possibly from primary source or secondary formation. Although, as the authors claimed, some compounds' source are unclear, cluster analysis, PCA or PMF are still useful to analyze the*

*source, even the source can be named as unknown source. At least we can understand what compounds are from the same or similar source.*

The objective of this study is not to perform a source apportionment (or source characterization) of organic compounds at the measurement site. This has already been done in Michoud et al. 2017 (ACP, 17, 8837–8865, https://doi.org/10.5194/acp-17-8837-2017, 2017) for this campaign. The objective here is rather to perform a description of the organic carbon composition in both phases at the molecular scale and to use this to study the partitioning behavior. Reference to main sources reported in the literature for the quantified compounds are mostly used, as an input, to explain the occurrence and behavior of these compounds at our measurement site, not, as an output, to perform a quantitative source apportionment as suggested by the first referee. Together with the reviewers we acknowledge that using molecular analysis to perform primary and secondary source apportionment would be really interesting. Nevertheless, we consider that this investigation could only come after the present discussion and this is why, we do not intend to perform PCA or PMF analysis on the data presented in the present paper. As it would enlarge significantly the revised manuscript which is not desirable and as it is by itself an ambitious reanalysis based on rather rare skills.

*Detailed comments:*
*Page 3, Line 19-20: TD-GCxGC-ToF/MS is the abbreviation of "Thermal desorption comprehensive two-dimensional gas chromatography–time of flight mass spectrometer". Notice that the expression of Gas ChromatographyxGC in line 20 is invalid.*
Modification has been done in the revised manuscript

*Page 4, Line 24: The expression of double chromatographic systems might be inappropriate and might be replaced by two-dimensional chromatography*
Modification has been done in the revised manuscript

*Page 6. Line 7- 9: All of these measurements were used to assess the composition of organic carbon and to estimate the experimental partition coefficient of compounds measured in both phases to be compared with theoretical values. The expression of this sentence seemed to suggest that the present manuscript displayed a large amount of data while the aim of which was just the partitioning coefficient estimation. As the last sentence of the introduction part, it is confusing why this study just focused on the partitioning of these compounds while their sources and mechanisms remained unknown.*
"All of these measurements" have been replaced by "These measurements" in the revised manuscript.
As already mentioned the present study does not only focus on partitioning coefficient since it also includes a study of the composition of organic carbon at the molecular scale and an estimation of the relative contribution to the total OA with an attempt of mass closure. Again, the objective of this study was not to perform another source apportionment which would be an interesting study but which cannot be considered as the ultimate result of atmospheric research on organic compounds.

*Page 9, Line 17: the diameter of the column (250 μm in Line 17) might be 0.25 mm as this unit was widely used.*
Modification has been done in the revised manuscript

*Page 9, Line 29: why pentadecane was utilized as the internal standard?*
The internal standard is used to assess the variability in mass spectrometer response or to identify a problem during the analysis. For this purpose we use a compounds with no derivatized

function. We chose pentadecane because of its low volatility which limits signal variability due to evaporation of the internal standard before the analysis. This latter information has been added to the revised manuscript.

*Page 19 Line 4: "than" to "as"*
Modification has been done in the revised manuscript

*Page 19 Line 9: It seems not reasonable that compounds in both gas phase and particle are high during day time. If it's controlled by the thermodynamic equilibrium, the compounds in the particle phase should be low during daytime due to evaporation.*
This part of the discussion concern the diurnal evolution of propanal and butanal in the gas phase only as it is stated at the beginning of the paragraph "Concerning the gaseous phase" (Page 19, line 3). To avoid misunderstanding the sentence has been modified as follow in the revised manuscript:
"During the campaign, these compounds in the gaseous phase are characterized by daily maxima during daytime and daily minima during nighttime"

*Page 19 Line 16-18, "For example…"unclear sentence*
This sentence has been removed in the revised manuscript and related references have been moved to the previous sentence.

*Page 20 Line 1: "observations are different than for that"*
Modification has been done in the revised manuscript

*There are plenty of wrong expression and gramma mistake. I list only a small part of them. I strongly suggest the authors to carefully go through the text.*
We went through the text and did our best to correct the mistakes in the revised manuscript.

**Response to referee #2:**

*I think that the manuscript could be published but after major revisions. I was also wondering whether an analytical chemistry journal might be more appropriate given that the methodological part is extensive. However, the study on partitioning coefficients makes it absolutely interesting and suitable for an atmospheric chemistry journal.*

We warmly thank Reviewer #2 for this general comment acknowledging the scientific interest of the multiphase measurements of organic compounds and their partitioning coefficient as a major by-product. We agree that in our will to carefully provide the technical basis support our rather novel techniques we may have given the impression of an analytical paper. We hope that a reorganization of the paper section may have provided more visibility to the discussions section.

*Full validation of the new method is not reported. Please add LOD, LOQ, precision and recovery for each compound in both matrices. Please add all data pertaining to line 1-6 on page 10 to the supplementary materials. Please add an example chromatogram showing the separation of the different compounds and please report all retention times.*
All these information have been added to the supplementary material inside the tables S2. A chromatogram has also been added in the supplementary material S3.

*Concerning the method used for the calculation of VOC concentrations, not all VOCs react with water cluster ion and those who react may react through a ligand-switching mechanism rather than proton transfer. The formula used for the calculations is not generally applicable to all VOCs and each VOC should be treated correctly (reaction with $H_3O^+$ only, reaction with both $H_3O^+$ and $H_5O_2^+$, proton transfer with $H_3O^+$ and ligand-switching with $H_5O_2^+$). At line 30 page 10, the authors refer to the signal of the $H_3O^+$ ion. Is this measured on the isotope $^{18}O$ and then corrected for the isotopic abundance? Was the $^{18}O$ used also for measuring the signal of $H_5O_2^+$?*

For all the compounds quantified by PTR-MS we used the signal at the protonated mass and we can therefore detect only the compounds that undergo proton transfer whether with $H_3O^+$ and/or with $H_3O^+(H_2O)$. Since we use direct calibration using standards for these compounds under similar instrumental conditions, the calibration factors take into account the relative contribution of each reaction pathways that lead to the ionization of the compounds by protonation as well as the fragmentation. The Xr factor allows also to take into account the relative contribution of protonation by reaction of the compounds with $H_3O^+(H_2O)$ as proposed by de Gouw and Warneke, 2007 (Mass. Spectrom. Rev., 26, 223–257, doi:10.1002/mas.20119, 2007). It is true that using this procedure does not allow to recover the concentrations of all the compounds that can be measured by PTR-MS.

For the signals of $H_3O^+$ and $H_3O^+(H_2O)$ we used isotope $^{18}O$ to avoid saturation of the signal and then corrected for the isotopic abundance. This information has been added in the revised manuscript as follow:

"$i_{H3O+}$ and $i_{H3O+(H2O)}$ the signals of $H_3O^+$ and $H_3O^+(H_2O)$ at m/z 19 and 37 respectively recorded at m/z 21 and 39 to avoid any saturation of the detector and recalculated using the isotopic ratio between $^{16}O$ and $^{18}O$".

*What was the extraction protocol for DNPH cartridges?*
This information has been added to the revised manuscript.

*Please add LOD, LOQ, precisions and recovery for the GC-MS method described at page 12-13.*
These information have been added to the revised manuscript.

*IC and HPLC-PAD analysis are mentioned but methods are not reported.*
IC (Ionic Chromatography) method is described in page 12 (lines 23-28). Mention to HPLC-PAD has been removed.

*Line 13 page 17 and elsewhere: may you be quantitative in discussing the agreement between methods rather than reporting qualitative adjectives such as "fair agreement"?*
Relative differences observed between the data have been added in the revised manuscript to add quantitative information. For the compounds experiencing the higher disagreements, the observed factors between measurements were already given and no relative differences are indicated.

*Line 31 page 17, was the thermodegradation observed with the standards?*
No thermodegradation was observed with the standards used.

*Line 9-16 page 18, did you check recovery and matrix effects to investigate the causes of this large disagreement between the results from the different methods?*
Recovery and matrix effects have not been investigated to explain the disagreement observed for malonic and tartaric acids.

*Concerning the section 3.2.5 on the partitioning of oxygenated compounds between the two phases, I have a general concern about the input data from the TD-GC-MS measurements. For some compounds, there was a clear disagreement between TDGC-MS results and alternative methods (e.g. glyoxal), so I am not confident that the data quality was always good enough to obtain reliable results on partitioning constants.*

It is true that some compounds, especially glyoxal, show clear disagreement between the different analytical methods used and that experimental partitioning coefficient for those compounds should be taken with care. However, even for these compounds the differences observed between theoretical and experimental partitioning coefficient cannot be explained neither by the overestimation of gas phase concentrations nor by the underestimation of particulate phase concentrations since we observe differences of several orders of magnitude (e.g. from 3 to 5 orders of magnitude for glyoxal) which is not the case in the intercomparison of measurements (1 order of magnitude for gas phase glyoxal).

This last point has been specified in the revised manuscript as follow:

"Furthermore, underestimation of gaseous concentrations for these compounds in such high proportion is unlikely, especially when we look at the comparisons performed for OVOCs with other measurement techniques (see section 3.2.2.1), even for compounds that shows strong disagreement between various analytical methods (e.g. glyoxal)."

*Concerning more specifically the calculations, it is mentioned that TSP concentrations are used but there was no mention on the paper about TSP measurements and all TD-GC-MS analysis were done on PM2.5. In equation 3, activity coefficients are used but there is no reference or table reporting which ones were used.*

It is true that no TSP measurements was performed and we used $PM_{10}$ mass concentrations measured by TEOM-FDMS instrument assuming that most of the TSP mass is contained in the $PM_{10}$ fraction at the measurement site. This has been specified in the revised manuscript. Activity coefficients was set to 1.27 as suggested by Rossignol, 2012

*Minor comments and typos:*
*The introduction is very long and could be more focused. For example, I would remove the paragraph from line 26 at page 3 to line 8 at page 4. It does not add much to the discussion.*

We propose to shorten this paragraph as follow in the revised manuscript:

"For this reason, analysis of principal component is often used to describe aerosol composition. Among them, Positive Matrix Factorization (PMF) applied to Aerosol Mass Spectrometer (AMS) spectra allows retrieving more information on the sources and nature of organic aerosol. Although this classification allows getting insight into the oxidation state of OA, it is not possible to identify chemical processes involved in SOA formation and aging."

*Line 6 page 7: did you check that the filter does not cause absorption of SVOCs?*

This is a possibility that has not been checked and there is no perfect solution on that matter. Without the use of a filter the risk is to trap compounds from the particulate phase in the cartridges causing overestimation, and with the use of a filter the risk is to lose some SVOCs by adsorption causing underestimation.

*Line 21 page 10: how humid was the humid zero air?*

Zero air generated this way is at the same relative humidity as ambient air as stated in line 22-23 of page 10.

*Line 18-20 page 12: why the calibration was performed only 2 months before and not on-site? Did you do any quality check? Why a collection efficiency of 0.5 was used rather than a composition-dependent one?*

ACSM measurements were performed continuously all over the year at this site and calibrations were performed regularly on-site but not during the specific period of the campaign.

On-site atmospheric concentrations delivered by the Q-ACSM were consistent during the entire campaign with NR-PM$_1$ and SO$_4$ concentrations obtained with colocated online instrument; scanning mobility particle sizer and a particle-into-liquid-sampler ion chromatograph (PILS-IC); see Fig. S3c of Michoud et al. (ACP, 2017).

ACSM observations were also consistent with co-located semi-continuous OCEC (Sunset Lab Field instruments), particle-into-liquid-sampler total organic carbon (PILS-TOC), TEOM-FDMS (PM$_1$), and 6-h time resolution (PM$_1$) filter sampling.

More information on the consistency of the ACSM results obtained during the campaign and using the collection efficiency of 0.5 can also be found in Arndt et al. (ACP, 2017), Chrit et al. (ACP, 2017; 2018).

*Line 18-20 page 13, the elution gradient is missing.*

The elution gradient has been added to the revised supplementary material S1.

*Line 23-25 page 15, how high the ratios? Was there any longrange transport of dusts that could have influenced these ratios?*

The ratio PM$_1$/PM$_{10}$ was comprised between 0.5 and 1 during this period and the ratio organic/PM$_{10}$ was comprised between 0.3 and 0.7. This has been specified in the revised manuscript as follow:

"A decrease of PM$_{10}$ concentrations is observed from 21 to 25 July (12.0 µg m$^{-3}$ on average) while the ratio PM$_1$/PM$_{10}$ and organic/PM$_1$ are the highest (comprised between 0.5 and 1 and 0.3 and 0.7 respectively)."

This period was characterized by air masses coming from north-east at the beginning and from western sector at the end. Long-range transport occurred therefore at the end of this period with air masses spending days above sea before reaching the measurement site but no transport of dusts occurred during this period.

*Line 14 page 21, are you referring to compounds that were in-between the LOD and LOQ?*

We are referring to compounds that are detected and/or quantified at least for some samples and are below LOQ for some samples. LOQ/2 are used to replace the value of these samples for total mass calculation.

*Line 29-30 page 21, I am not sure if the discussion around Henry's law constants is relevant here. The absolute solubility of a compound in water may be more relevant in this case if we are discussing a transfer from particulate into water rather than a gas-water equilibrium.*

In this section, we are discussing the transfer from gas phase to deliquescent aerosol and Henry's law constant is therefore relevant here.

*Line 4 page 25, oligomer production can happen also at high relative humidity and in the aqueous phase.*

Clarifications have been made in the revised manuscript as follow:

"These reactions are favored under low water content in the particles even though oligomer production from other reactions can also happen at high relative humidity and in the aqueous phase."

*Line 27 page 25, I believe "emphasis" should be "emphasizes" in this sentence. English is generally okay although it could be improved in some places.*

The proposed correction has been performed in the revised manuscript. As answered to the first reviewer, we went through the text and did our best to correct the mistakes in the revised manuscript.

*Please translate from French into English all labels in the plots in supplementary 2.*

The proposed correction has been performed in the revised supplementary.

*In the table in supplementary 1 please change "substitut" with "substitute".*

The proposed correction has been performed in the revised supplementary.

---

## Author Response (AR2)

First of all, we would like to thank again the reviewers for their comments on the manuscript. The changes made to the revised manuscript are summarized below.

**Response to referee #1:**

*I believe the authors response to most of my comments. The technique this work deployed is advanced, and the data are beautiful. However, my only concern is that the current manuscript is, not precisely, in between of a research paper and a measurement report paper. The data analysis is covered by the techniques. It's far beyond a analytical chemistry paper, but still defective as an atmospheric chemistry paper. If the authors insist to submit it as a research paper, I have to say no with regret.*

We certainly insist to publish it as a research paper. However, to take into account the remarks of the referee #1 and as suggested by the editor, we shortened the technical section 2 moving important part of the technical description and details (about 6.5 pages and 2 tables) into the supplementary material (S1, S2 and S3). The technical section is now 1.5 pages which is more reasonable and makes the article more balanced with the result section being 4.5 pages (plus 1 table and 8 figures) and the discussion section being 7 pages (plus 1 table and 5 figures).

**Response to referee #2:**

*1. Concerning the quantification of VOCs with a PTR-MS, I am not entirely sure I understand the answer provided by the authors. From the manuscript I understand that for each VOC both the $H_3O^+$ and $H_3O^+(H_2O)$ signals were used for quantification. If this is the case then my comment has not been addressed. While the formula used for VOC quantification is correct (I have never questioned that), its application must change depending on the relevant proton transfer reactions:*

*- for VOCs that react with $H_3O^+$ only and do not react with the water cluster, only the $H_3O^+$ signal should be used for the quantification because the $VOC+H^+$ signal will depend only on the proton transfer reaction with $H_3O^+$*

*- for VOCs that react with $H_3O^+$ and with $H_3O^+(H_2O)$ via proton transfer, then both the $H_3O^+$ and $H_3O^+(H_2O)$ signals should be used for the quantification*

*- for VOCs that react with $H_3O^+$ via proton transfer reaction and with $H_3O^+(H_2O)$ via ligand switching reaction, the product ion will appear at two different masses corresponding to $VOC+H^+$ and $VOC+H_3O^+$. If the author decides to base the quantification only on the $VOC+H^+$ signal, which is produced by the main reaction pathway, then only with $H_3O^+$ signal should be used for the quantification because the reaction between the VOC and $H_3O^+(H_2O)$ does not produce the $VOC+H^+$.*

We agree with the reviewer that the normalization procedure used for VOC quantification should not be the same for all VOCs and be dependent of the way it reacts with the reagent (i.e. $H_3O^+$ and/or with $H_3O^+(H_2O)$). This is actually the case in our quantification method since a factor (Xr, see eq. 1), which is compound dependent, is applied to the signal of $H_3O^+(H_2O)$. This factor is determined experimentally for each quantified VOCs and allows to take into account the different reaction pathways leading to the protonation of each VOC. In other word, if a VOC does not react with $H_3O^+(H_2O)$ or if its reaction with $H_3O^+(H_2O)$ proceed via ligand switching rather than by proton transfer, this factor will be zero and quantification will only be

performed normalizing the signal by $H_3O^+$. This has been clarified in the revised supplement S2 where the description of PTR-MS data acquisition protocol has been moved (see response to referee #1):

"Xr is a factor introduced to account for the effect of humidity on the PTR-MS sensitivity (de Gouw and Warneke, 2007) and is determined experimentally through calibrations performed at various relative humidity for each individual quantified VOCs. It therefore also allows to take into account the reaction pathways of each individual VOC with the reagent ions (i.e. $H_3O^+$ and/or with $H_3O^+(H_2O)$)."

*2. The evaluation of recovery and matrix effects should be part of the validation of any new analytical protocol. Considering that there were discrepancies between the results obtained with different methods I strongly recommend completing the method validation and add this information to the manuscript.*

Actually, recovery and matrix effects have been evaluated for our analytical method during its development phase. These optimization and evaluation tests have been conducted introducing known amount of standards in an atmospheric simulation chambers (CESAM, Wang et al., 2011) and a laboratory test chamber (Gonzalez-Flesca and Frezier, 2005) for various humidity conditions and complexity of mixtures and are discussed by Rossignol et al. (2012).

In addition, recovery and matrix effects are taken into account in the way we perform the quantification in our analytical method. For quantification, we use external calibrations performed in the same conditions as atmospheric samples, doping the collecting support (i.e. whether filter or adsorbent cartridges) with known amount of standards. This way, response coefficient used for quantification include the derivatization and extraction efficiency and the matrix effects.

Our answer to the previous comment of the referee #2, stated that we did not investigated this specifically to explain the disagreement observed for malonic and tartaric acids, which was the original question of the referee #2. We apologize for the misunderstanding. We add a statement to clarify this in the revised supplement S1 where the internal and external calibration protocol of TD-GC-MS analysis has been moved (see response to referee #1):

"In addition of this internal calibration protocol, external calibrations are performed in the same conditions as atmospheric samples, doping the collecting support (i.e. whether filter or adsorbent cartridges) with known amount of external standards (list of external standards can be found in the supplementary material 4). This way, response coefficient used for quantification include the extraction and derivatization efficiency and the matrix effects. In addition, recovery and matrix effect evaluation for the method can be found in Rossignol et al. (2012)"

*3. Please add to the manuscript the information about the activity coefficients used for equation 3. I see the information in the answers to the reviewers, but I haven't spotted it in the manuscript (unless it is there, and I missed it).*

This information has been added in the revised manuscript, as follow:
"$\zeta_i$ has been set to 1.27 as suggested by Rossignol (2012)"

*4. Concerning my comment on page 21 about the Henry's law constant, please add to the manuscript that you are discussing the transfer from gas phase to deliquescent aerosol. It was not obvious to me, I thought you were discussing the transfer of analytes from the particles to liquid water inside the PILS.*

This information has been added in the revised manuscript, as follow:
"Some of the compounds identified and quantified by TD-GC/MS, especially carboxylic acids, are soluble in aqueous phase and their presence in aerosol phase could proceed through the transfer from gas phase to deliquescent aerosol."

**References**

de Gouw, J. and Warneke, C.: Measurements of volatile organic compounds in the earth's atmosphere using proton-transferreaction mass spectrometry, Mass. Spectrom. Rev., 26, 223–257, doi:10.1002/mas.20119, 2007.

Gonzalez-Flesca, N. and Frezier, A.: A new laboratory test chamber for the determination of diffusive sampler uptake rates, Atmos. Environ., 39, 4049–4056, doi:10.1016/j.atmosenv.2005.03.025, 2005.

Rossignol, S.: Développement d'une méthode de prélèvement simultané et d'analyse chimique des phases gazeuse et particulaire atmosphériques pour une approche multiphasique de l'aérosol organique secondaire, Paris 7. [online] Available from: http://www.theses.fr/2012PA077208 (Accessed 14 February 2016), 2012.

Rossignol, S., Chiappini, L., Perraudin, E., Rio, C., Fable, S., Valorso, R. and Doussin, J. F.: Development of a parallel sampling and analysis method for the elucidation of gas/particle partitioning of oxygenated semi-volatile organics: a limonene ozonolysis study, Atmospheric Meas. Tech., 5(6), 1459–1489, doi:10.5194/amt-5-1459-2012, 2012.

Wang, J., Doussin, J. F., Perrier, S., Perraudin, E., Katrib, Y., Pangui, E., and Picquet-Varrault, B.: Design of a new multi-phase experimental simulation chamber for atmospheric photosmog, aerosol and cloud chemistry research, Atmos. Meas. Tech., 4, 2465–2494, doi:10.5194/amt-4-2465-2011, 2011.